# Evaluation of information flows in the RAS-MAPK system using transfer entropy measurements

**Nobuhisa Umeki[1], Yoshiyuki Kabashima[2,3], Yasushi Sako[1]***

[1]Cellular Informatics Laboratory, RIKEN, Cluster for Pioneering Research, Wako, Japan; [2]Institute for Physics of Intelligence, The University of Tokyo, Bunkyo-ku, Japan; [3]Trans-Scale Quantum Science Institute, The University of Tokyo, Bunkyo-ku, Japan

**Abstract** The RAS-MAPK system plays an important role in regulating various cellular processes, including growth, differentiation, apoptosis, and transformation. Dysregulation of this system has been implicated in genetic diseases and cancers affecting diverse tissues. To better understand the regulation of this system, we employed information flow analysis based on transfer entropy (TE) between the activation dynamics of two key elements in cells stimulated with EGF: SOS, a guanine nucleotide exchanger for the small GTPase RAS, and RAF, a RAS effector serine/threonine kinase. TE analysis allows for model-free assessment of the timing, direction, and strength of the information flow regulating the system response. We detected significant amounts of TE in both directions between SOS and RAF, indicating feedback regulation. Importantly, the amount of TE did not simply follow the input dose or the intensity of the causal reaction, demonstrating the uniqueness of TE. TE analysis proposed regulatory networks containing multiple tracks and feedback loops and revealed temporal switching in the reaction pathway primarily responsible for reaction control. This proposal was confirmed by the effects of an MEK inhibitor on TE. Furthermore, TE analysis identified the functional disorder of a SOS mutation associated with Noonan syndrome, a human genetic disease, of which the pathogenic mechanism has not been precisely known yet. TE assessment holds significant promise as a model-free analysis method of reaction networks in molecular pharmacology and pathology.

*For correspondence:
sako@riken.jp

**Competing interest:** The authors declare that no competing interests exist.

## Editor's evaluation

Intracellular signaling pathways and reaction networks are conventionally modeled analytically using ordinary differential equations. This paper demonstrates nicely how a statistical approach, transfer entropy, can be applied for model-free assessment of the timing, direction, and strength of the information flow regulating a system. Specifically, transfer entropy is applied to time-lapse imaging data to evaluate feedback loops in the EGFR-SOS-RAS-RAF pathway and the effects of drugs and mutations. Overall, the paper will be important for a wide audience including systems biologists and molecular pharmacologists, and is supported by compelling evidence.

## Introduction

Assessing the functions of intracellular reaction networks is a central focus of contemporary molecular cell biology. Significant progress has been made through biochemical and genomic analyses, revealing the networks' elements and their interconnections. Analytical models based on coupled ordinary differential equations (ODEs) are widely used to explore the most probable network structure

and determine reaction parameter values. These models offer mechanistic insights into the network and facilitate predictions of molecular activation dynamics (*Omony, 2014*; *Tavassoly et al., 2018*). However, due to the networks' molecular diversity and structural complexity, the credibility of analytical models remains partially uncertain in many cases. It would be beneficial to have a method to confirm the results of ODE modeling from a different perspective. In addition, it may be important to note that the average concentration of activated molecules estimated in ODE models does not necessarily imply the ability to control downstream molecules. Control capability refers to how differences in the upstream components are faithfully read out by the downstream in the presence of large noises in the reactions and environments. Unfortunately, ODE models typically do not account for reaction fluctuations.

Statistical models complement analytical ones (*Waltermann and Klipp, 2011*). Statistical analyses are fundamentally model-free, making them robust when information about the network structure and reaction formulas is limited. Results obtained from information analyses based on statistics are expressed in universal units, such as bits or nats, making them suitable for comparison. These units are independent of the specific properties and functions of individual molecular elements within the network. Mutual information (MI) and transfer entropy (TE) are examples of measures used in information analyses (*Schreiber, 2000*; *Gencaga et al., 2015*). TE, in particular, possesses the power to quantify the information flow, which reflects the causal relationships between different reaction time courses (*Granger, 1969*; *Barnett et al., 2009*). When estimating the future intensity of the reaction from its past intensity, the estimation accuracy could be improved by referring to the past intensity of another reaction. TE implies this improvement. (See Supplementary text for a more detailed explanation.) Since TE primarily means the control capability of upstream behavior on downstream components, the TE time course provides insights into the dynamics of signal transduction from the downstream effect perspective.

In practice, although MI analysis has found various applications in examining snapshots of cell signaling reactions (*Uda et al., 2013*; *Miyagi et al., 2021*; *Fuchsberger et al., 2023*), the use of TE assessment for studying the dynamics of real intracellular reaction networks has been significantly limited up to the present time. This is primarily because conventional TE calculation methods require extensive simultaneous data sequences and substantial computational power. The use of TE in biology is generally limited to simulation studies (*Pahle et al., 2008*; *Kamberaj and van der Vaart, 2009*; *Ursino et al., 2020*) and neural response analysis, which have high reproducibility and low noise (*Wibral et al., 2011*; *Stetter et al., 2012*). There have been only a few applications of TE to study noisy intracellular chemical reactions (*Pahle et al., 2008*; *Ito and Sagawa, 2015*). However, we recently have reported a method for TE assessment applicable to relatively small numbers of time sequences (*Imaizumi et al., 2022*). This new approach involves utilizing a Gaussian approximation of the response distributions, enabling the rapid evaluation of TE using covariance matrices (*Appendix 1—figure 1*). Additionally, we have developed a method for result evaluation using bootstrapping. By applying these techniques, we detected significant levels of TE between SOS and RAF proteins following epidermal growth factor (EGF) stimulation, utilizing several hundred single-cell simultaneous response time courses.

SOS and RAF are essential components of the EGFR-RAF-MAPK system, which plays a crucial role in regulating cell fate (*Lemmon and Schlessinger, 2010*). Upon activation of the EGF receptor (EGFR), tyrosine phosphorylations occur in its cytoplasmic domain, leading to the translocation of a protein complex comprising GRB2 and SOS from the cytoplasm to the plasma membrane, where they recognize the EGFR phosphorylations. SOS functions as a guanine-nucleotide exchanger for RAS, a small GTPase located on the cytoplasmic side of the plasma membrane. The translocation of SOS induces the activation of RAS, which, in turn, results in the translocation of RAF, a serine/threonine kinase and an effector of RAS, from the cytoplasm to the plasma membrane. These intracellular translocations of SOS and RAF can be visualized using a total internal fluorescence microscope with fluorescently labeled SOS and RAF molecules, specifically observing near the basal surface of cells (*Nakamura et al., 2017*; *Yoshizawa et al., 2021*).

Given the significance of RAS as a key regulator of various cellular responses, understanding the signal transduction from SOS to RAF, from the upstream to downstream events involved with RAS activation, is of interest. Human cancer (*Cai et al., 2019*) and genetic diseases, such as Noonan syndrome (*Aoki and Matsubara, 2013*; *Aoki et al., 2016*), involve various mutations in SOS, which are expected

to lead to the hyperactivation of the RAS-MAPK system. However, the specific molecular mechanisms underlying these SOS mutations remain largely unknown. In a previous study (*Imaizumi et al., 2022*), we observed that one of the Noonan mutations of SOS (R1131K) affected the information flows from RAF to SOS. This finding raised the possibility that TE between SOS and RAF might provide insights into the functions of the RAS-MAPK system, yet detailed analysis is lacking. In the present study, we aim to investigate the properties of TE between SOS and RAF more precisely to demonstrate the potential of TE analysis for studying intracellular reaction networks and gain deeper insights into SOS/RAF signaling dynamics.

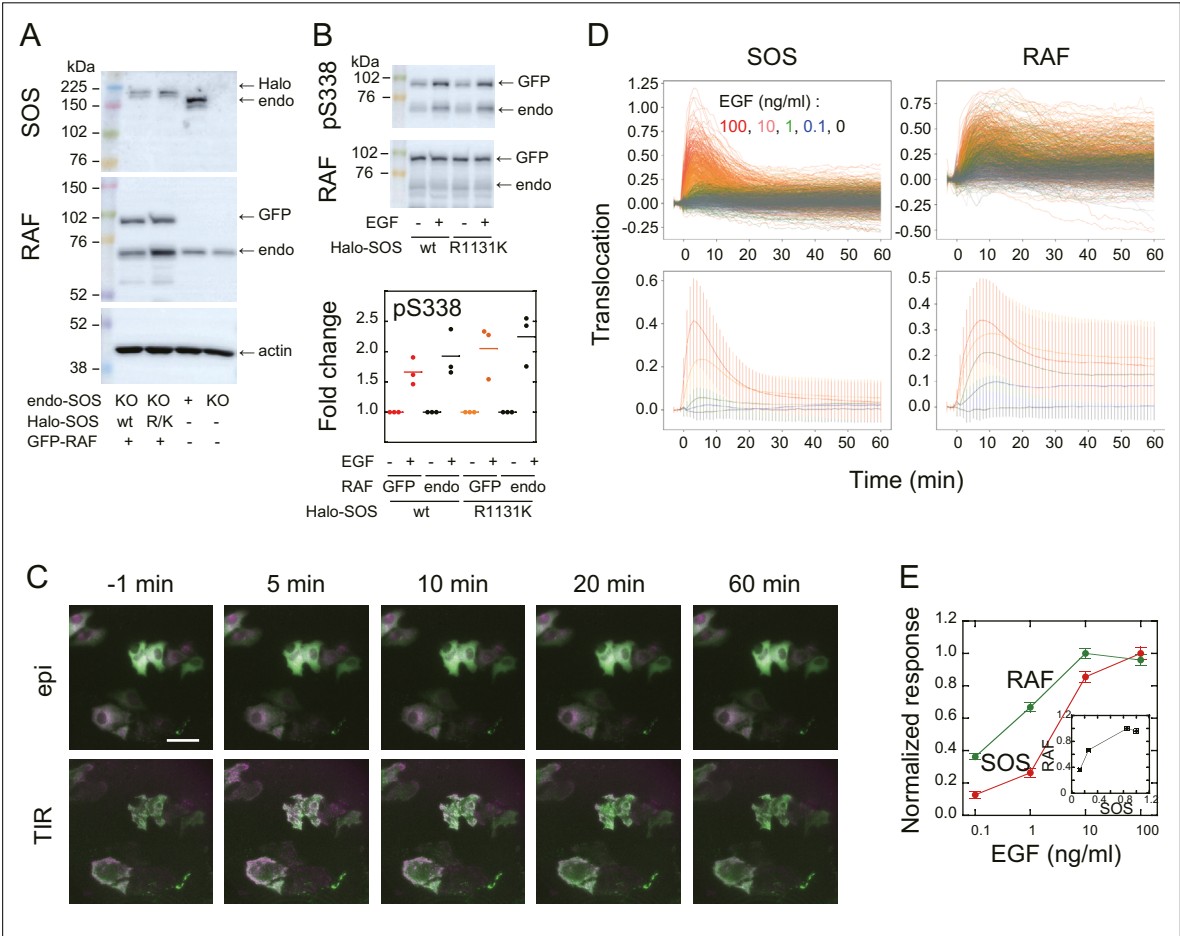

**Figure 1.** SOS and RAF responses to cell stimulation. (**A**) Expression levels of SOS and RAF probes. Halo-SOS (wild type, wt or R1131K mutant, R/K) and GFP-RAF were expressed in SOS knockout HeLa cells. (**B**) RAF phosphorylation before and after 10 min of 10 ng/ml EGF stimulation of cells. Results of representative Western blotting and quantification of three independent experiments. (**C**) Translocations of SOS (magenta) and RAF (green). Epi-fluorescence (upper) and TIR-fluorescence (lower) images at the indicated times after 100 ng/ml EGF stimulation. Bar, 50 μm (*Appendix 1—Videos 1 and 2*). (**D**) Time course analysis of SOS and RAF translocation observed in the total 1467 cells after vehicle (black: 150 cells) or EGF (blue: 0.1 ng/ml, 300 cells; green: 1 ng/ml, 310 cells; orange: 10 ng/ml, 370 cells; red: 100 ng/ml, 337 cells) stimulation. Changes in the molecular density on the cell surface (TIR) relative to the whole cell signal (epi) are plotted. The lower panels show the average with SD of the single cell data in the upper panels. (**E**) EGF dose dependence of the SOS (red) and RAF (green) responses. Response intensities were determined as the sum of the molecular translocation during 0–60 min normalized to the maximum. Inset shows the relationship between SOS and RAF responses. Cell-to-cell averages are shown with SE.

The online version of this article includes the following source data for figure 1:

**Source data 1.** Source data for *Figure 1A*.

**Source data 2.** Source data for *Figure 1B*.

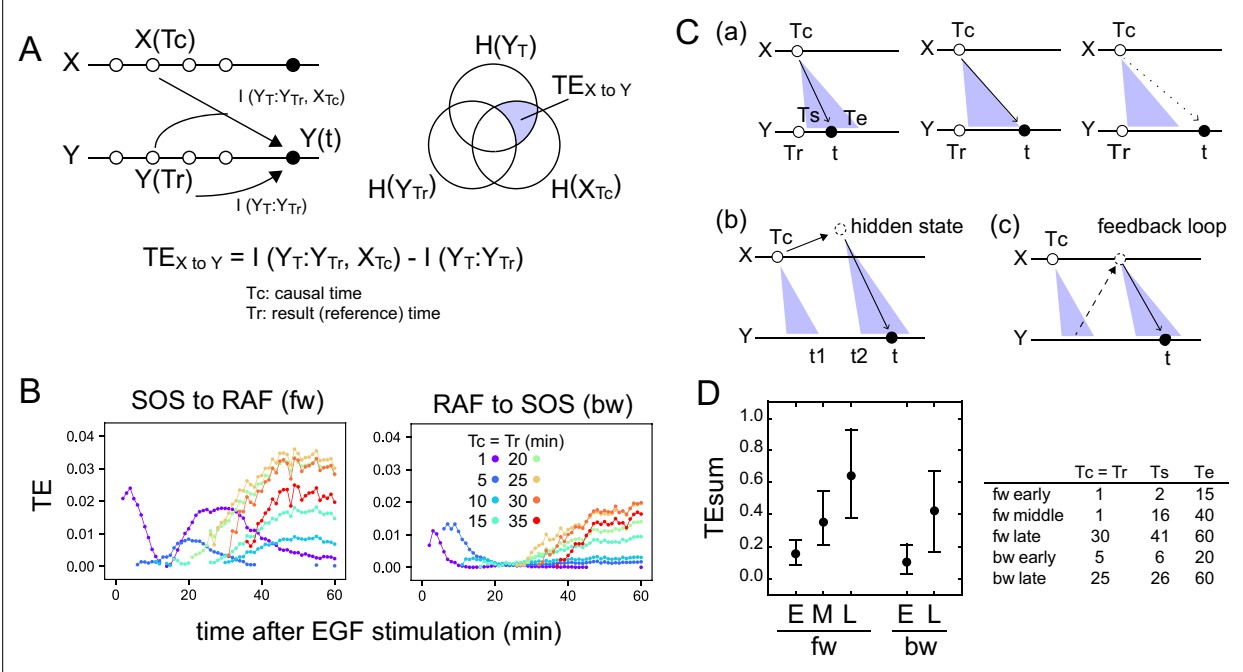

**Figure 2.** Transfer entropy time course analysis. (**A**) Concept of TE between two reaction time courses. TE from X(Tc) to Y($t$) was calculated using Y(Tr) as the reference (Tc = Tr < $t$). (**B**) TE time courses for the total 1317 pairs of translocation trajectories with 0.1–100 ng/ml of EGF stimulation. Calculations were performed for the indicated Tc = Tr from 1 to 35 min after cell stimulation. Only statistically significant TE values (see Materials and methods) are plotted. (**C**) Appearance of the separated TE peaks from single Tc. (**a**) In a single-track reaction pathway, TE from Tc in the X time series appears during a single continuous period (Ts to Te) and disappears after $t \geq$ Te. (**b**) The second TE peak is indirect information flow through a hidden state outside the pathway of the first TE peak. (**c**) The second TE peak can be caused by a feedback loop. (**D**) The sum of the significant TE (TEsum) in the TE peaks. Average and 5–95% range in 100-time bootstrapping are shown. TE from Tc (=Tr) was integrated during $t_s \sim t_e$ as TEsum. Tc, $t_s$, and $t_e$ used in the calculations are shown in the list. E: early, M: middle, L: late.

## Results

### Simultaneous measurement of SOS and RAF responses in single cells

To visualize intracellular translocations of SOS and RAF subtypes, hSOS1 and CRAF, hSOS1-Halo and GFP-CRAF (hereafter SOS and RAF, respectively) were expressed in HeLa cells, in which the intrinsic wild-type hSOS1 gene was knockout (*Figure 1A*). Upon EGF stimulation, the phosphorylation of both intrinsic and GFP-fused RAF at S338 increased, indicating their activation (*Figure 1B*). After the tetramethylrhodamine (TMR)-ligand conjugation to Halo fused to SOS, cells were stimulated with EGF, and changes in fluorescent SOS and RAF densities on the basal plasma membrane were quantified simultaneously in single cells through epi and total internal reflection (TIR) fluorescence microscopies (*Figure 1C*; *Appendix 1—Video 1* and *Appendix 1—Video 2*). The intensities of translocation responses varied with the EGF dose (ranging from 0 to 100 ng/ml; *Figure 1D*). To obtain the full data set, six independent measurements were performed for each EGF dose. For most measurements, the variation in daily means was less than 30% of the total cellular response range (*Appendix 1—figure 2*). No significant translocation of SOS and RAF was observed in cells following vehicle stimulation (*Figure 1C* black and *Appendix 1—figure 3A*). To quantify SOS and RAF responses, the integration of response time profiles during 0–60 min of EGF stimulation was measured (*Figure 1E*). The RAF response exhibited higher saturation with varying EGF doses compared to that of SOS.

### Transfer entropy between SOS and RAF dynamics

The TE between the response sequence of SOS and RAF was evaluated both in the directions from SOS to RAF (forward; fw) and from RAF to SOS (backward; bw). In the fw (bw) TE calculation, we refer to the SOS (RAF) sequence as the causal sequence, X($t$), and the RAF (SOS) sequence as the result sequence, Y($t$) (*Figure 2A*). The TE from a fixed causal time point (Tc) to the result sequence was calculated along $t$, concerning the distribution of Y at a fixed reference time point (Tr). In this study,

TE calculation was performed for Tc = Tr < t. The 3D distribution of SOS and RAF at Tr, Tc, and $t$ was unimodal and fairly symmetric in its main bodies (*Appendix 1—figure 4*), allowing the Gaussian distribution as a first approximation.

TE time courses were calculated varying Tc (=Tr) from 1 to 35 min, using all measurements at EGF concentrations of 0.1–100 ng/ml (*Figure 2B*). Statistically significant levels of TE were detected in both fw and bw directions. Statistically insignificant TE values were close to zero in our cases and were removed from the TE time course plots. Values of MI between X(Tr) and X($t$) and Y(Tc) and X($t$) were statistically significant for all time points Tc (=Tr) and $t$, and their significance thresholds were smaller than the TE values (*Appendix 1—figure 5*). The detections of TE were not due to the limited number of data sequences, and the bootstrap mean values of TE were stable to the data number (*Appendix 1—figure 6*). TE values obtained after vehicle stimulation were one order of magnitude lower than those obtained after EGF stimulation (*Appendix 1—figure 2B*). The detection of significant levels of bw-TE suggests the presence of at least one feedback loop from RAF with a significant ability to control the SOS response and/or the presence of a common input to SOS and RAF. (See 'Discussion' and 'Stochastic ODE simulations' in the supplement for this point.)

The time courses of TE show that TE from a Tc in the early stages of the cell response was transmitted as multiple peaks. For example, in the fw direction, TE from Tc = 1 min formed two peaks at $t$=4 and 25–35 min. Between these two peaks (t~15 min), fw-TE almost disappeared. Also in the bw direction, TE from Tc = 1–10 min formed two peaks before 20 min and after 30 min. The second peaks were not large but statistically significant. The values of bw-TE were insignificant or small during $t$=20–30 min. Such separation of TE peaks from a single time point implies indirect information transfer resulting in the second peak (*Figure 2C*). In a reaction cascade without branches, information from the upstream reaction is conveyed to the downstream reaction as a single peak (*Figure 2Ca*). No TE should be observed after the first information flow has ended ($t$>Te). Multiple TE peaks indicate that the information from Tc was once transferred to some hidden states and passed to the late periods in the result sequence as the later peaks (*Figure 2Cb*). During the period between the end of the first peak and the beginning of the second peak from the hidden state ($t = t_1–t_2$), TE disappears. The hidden states can be generated through a feedback loop (*Figure 2Cc*). The exchanges of fw- and bw-TE between SOS and RAF were observed in practice. The fw (or bw)-TE from the peak time of the previous bw (or fw)-TE time course generated a peak from which the next round of fw (or bw)-TE peaks were generated (*Appendix 1—figure 7*). This finding supports that at least a portion of TE was alternatively transferred between SOS and RAF using a feedback loop.

In our calculation, at least three and two TE peaks were observed at different periods in the fw and bw directions, respectively. We named the three fw peaks as early, middle, and late fw peaks, and the two bw peaks as early and late bw peaks. To briefly capture the characteristics of the multiple-peak TE time courses, we use the sum of TE amounts (TEsum) during the periods of TE peaks in the typical calculation conditions. The representative Tc (=Tr) and the range of $t$ for each peak used in the evaluations of TEsum are shown in *Figure 2D*. The TEsum values were estimated as the average of 100 bootstrap runs. Although the accuracy of TEsum estimation depends on the number of data sets, estimation errors can be reduced by taking the average of bootstrapping (*Appendix 1—figure 6B*). The average converged to nearly identical values (<4% differences) for >500 data sets.

## Dependency on the EGF dose

We examined the EGF dose dependencies of the SOS and RAS response intensities, as well as the TEsum values resulting from these responses, by analyzing the reaction time courses obtained under varying EGF concentrations (*Figure 1D*). The intensity of the SOS and RAF responses was defined as the integration of the molecular translocation during the periods of TE peaks, as used for the TEsum calculations. The response intensities at the early fw- and bw-peaks exhibited saturable and sigmoidal profiles to the EGF dose in a log scale, respectively, as is commonly observed for intracellular reactions (*Figure 3A*). The dose dependencies at the later peaks were bell-shaped with a faint peak at 10 ng/ml EGF.

The time course of TE for each EGF dose examined consisted of multiple peaks (*Appendix 1—figure 8*) as observed in the calculations using the mixed data (*Figure 2B*). The TEsum values calculated at each peak in the TE time courses showed a sharp bell-shaped EGF dose dependency in every case (*Figure 3B*). This is in contrast to the EGF dose dependencies of the response intensities

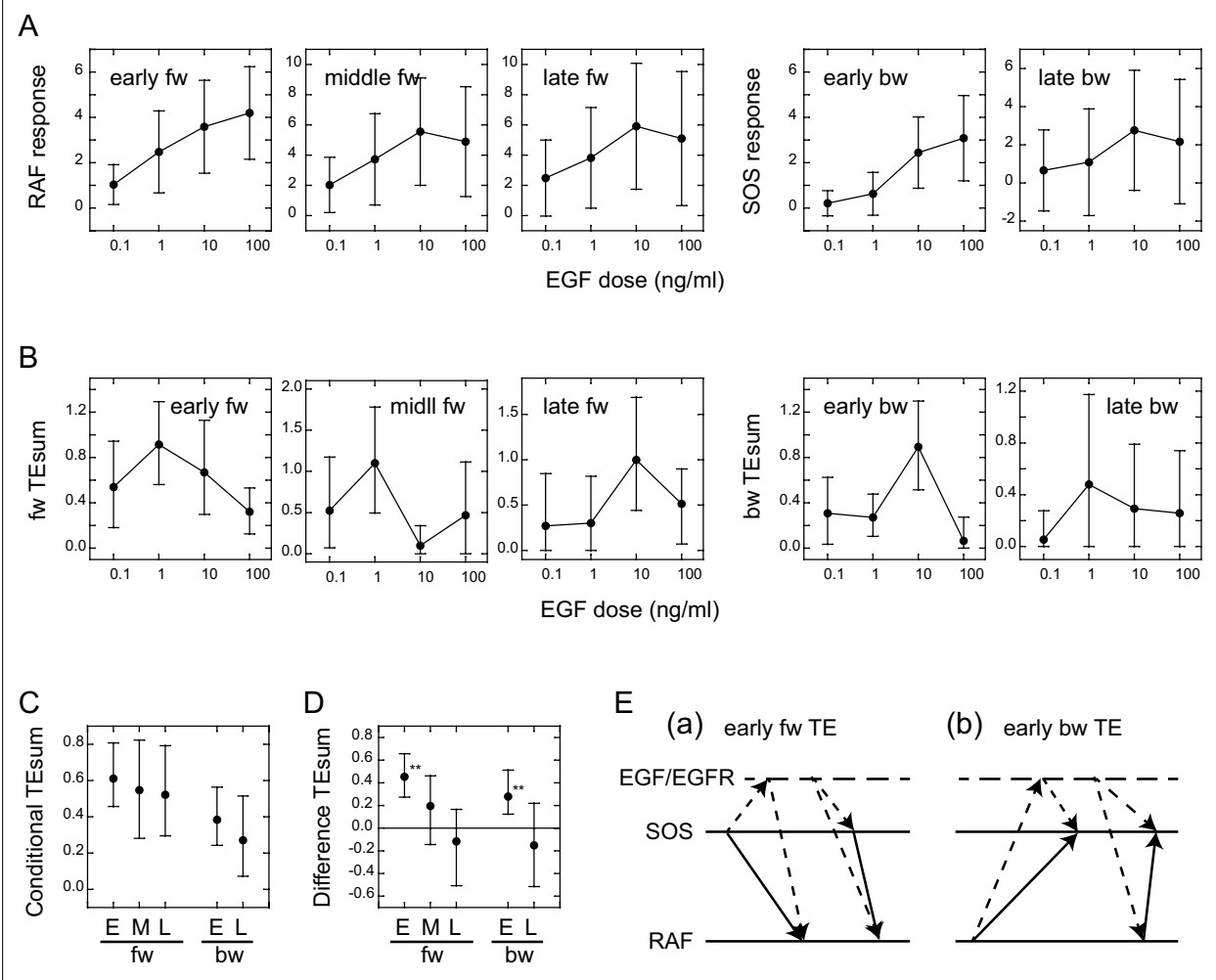

**Figure 3.** Conditional transfer entropy analysis to the EGF dose. (**A**) EGF dose dependency of SOS and RAF responses at the TE peaks. Response time courses (*Figure 1D*) were integrated during the TE peak periods listed in *Figure 2D*. The averages for single cells were plotted with SD. (**B**) EGF dose dependency of TEsum for each TE peak. (**C**) TEsum conditioned to the EGF dose. (**D**) Difference between TEsums conditioned (**C**) and non-conditioned (*Figure 2D*) to the EGF dose. **bootstrapping p<0.05. (B–D) The averages obtained from 100-time bootstrapping are shown with the 5–95% interval. (**E**) Multiple TE tracks involving EGF/EGFR complex suggested form (**D**) (see text for details). Two possible cases are presented for both early fw-TE (**a**) and early bw-TE (**b**) peaks.

(*Figure 3A*). The maximum values in the TEsum were often observed at the EGF doses distinct from those that elicited the maximum response intensity, even though the response and TE originated from the same causal reaction. This result shows that TE analysis provides different information than what is obtained through typical reaction response analyses.

To investigate the effect of EGF on TE between SOS and RAF, we compared the TEsum amounts conditioned and non-conditioned to the EGF dose (*Vakorin et al., 2009*). First, TEsum was calculated for each EGF dose, and then the conditional TEsum was calculated as the average of the TEsum values at different doses (*Figure 3C*). The TEsum calculated using the mixed data under all EGF concentrations represents the non-conditional TEsum (*Figure 2D*). To evaluate the significance of differences between conditional and non-conditional TEsums, we conducted 100 time bootstrapping trials, calculated the difference between the TEsum for each random data sampling, and obtained the average and 5–95% confidence range (*Figure 3D*). Statistically significant differences were observed in the early fw- and bw-peaks. In these peaks, conditional TEsum exceeded non-conditional TEsum. This result was unexpected because, in the expected pathway of the tandem reaction from the EGF/EGFR complex to RAF via SOS, the non-conditional TE that includes the uncertainty of the most upstream element (EGF) should be equal to or greater than the TE that is conditioned to EGF, which mostly

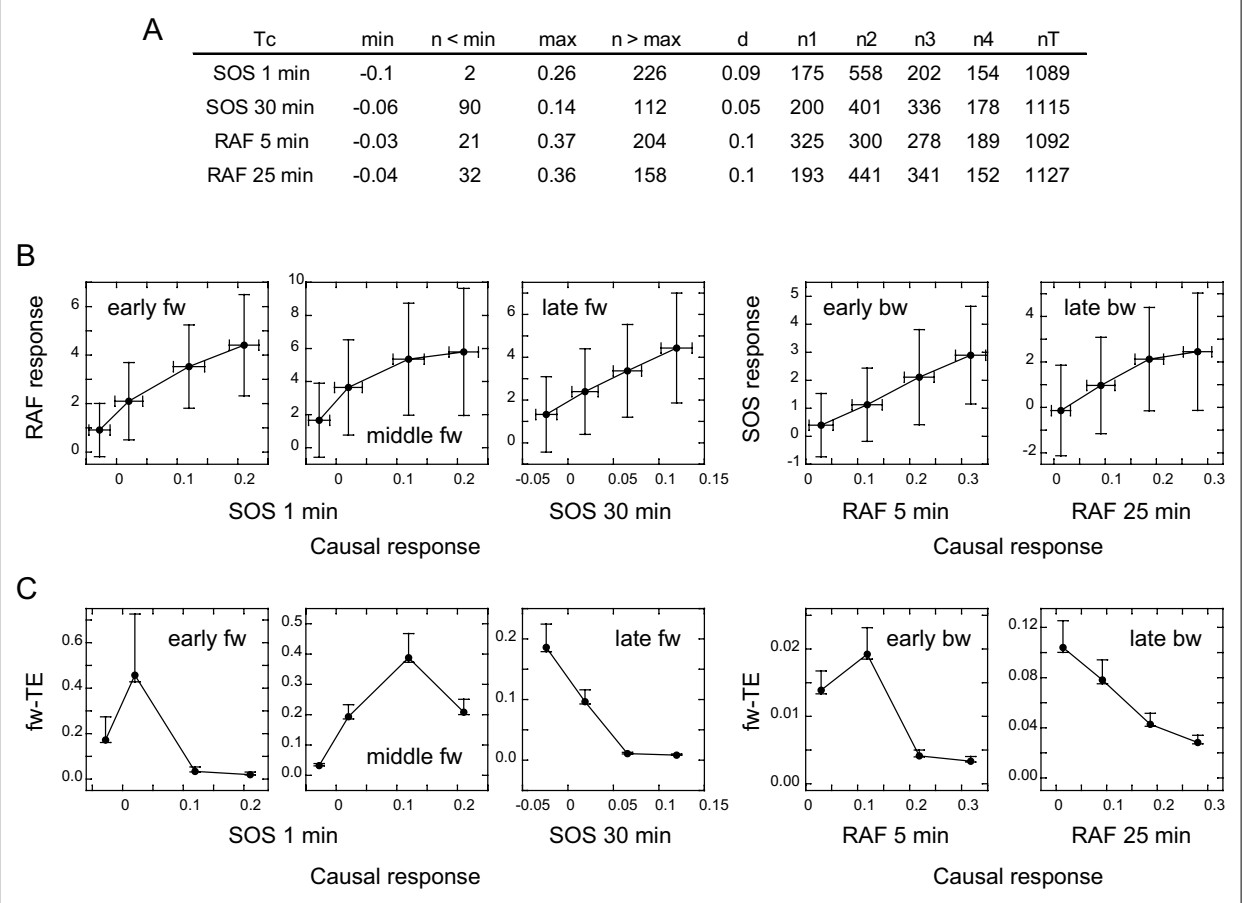

**Figure 4.** Response/transfer entropy relationships. Four groups of single-cell response time courses were extracted from the total data sets (*Figure 1D*) according to the causal response intensity at Tc. (**A**) Parameter list for data extraction. Each group had fixed bin sizes with >150 data sets. Data in the low and high ends of distributions were not used. min, max: the minimum and maximum response intensities, respectively. n<min, n>max: the excluded data numbers in the low and high end for calculation, respectively. d: bin size. n1–n4: data numbers contained in each bin. nT: the total data number. (**B**) Relationships between the SOS and RAF responses. The response intensities during the period of TE peaks in the result sequence are plotted as a function of the causal response intensity at Tc. Cell-to-cell averages are plotted with SD. (**C**) Relationships between the causal response intensity and the corresponding TEsum. Averages from 100-time bootstrapping are plotted. Error bars show the estimated error range for the bootstrap average for n>150 (*Appendix 1—figure 2*).

rejects the uncertainty of the EGF dose (Supplement text 'TE and conditional TE'). The larger conditional TEsum suggests that there was a bypass involving EGF/EGFR in the regulation of RAF response outside the well-known EGFR/SOS/RAF pathway (*Figure 3Ea* left and Supplement text 'Stochastic ODE simulations and *Appendix 1—figure 11*). Similarly, the SOS response was regulated by EGF/EGFR outside of the bw pathway from RAF that did not involve EGF/EGFR (*Figure 3Eb* left). These bypasses mainly functioned in the early phase (<15–20 min) of EGF signaling. Another possibility is that SOS and RAF were regulated by EGF/EGFR through parallel pathways, causing a pseudo-correlation between the SOS and RAF reaction time courses (right in *Figure 3Ea, b*).

## Response/transfer entropy relationships

TE in our calculation was derived from the SOS (for fw-TE) or RAF (for bw-TE) response at Tc after EGF stimulation. To investigate the correlation between the intensity of the SOS or RAF response and the amount of TE directly, we classified the reaction time courses according to the response intensity at Tc. The classification used the same bin size to avoid the effect of the dynamic range in the causal response. As a result, the number of data sets was not identical for each response class. To ensure reasonable numbers (>150) of data sets for the calculation, data in the larger and smaller ends of the response distributions were not used (*Figure 4A*). We then examined the response (*Figure 4B*) and

TE (*Figure 4C*) profiles against the causal response intensity. We estimated the error range for the bootstrapped mean of TEsum from the calculations using different numbers of random samples from all datasets (*Appendix 1—figure 6B*), and expected the means to be within 63–107% (early fw peak) or 83–104% (other peaks) range of the true values, when >150 data sets were used.

The response intensity in the result sequence was either linearly or saturatedly increased with the response intensity at the causal time responsible for it (*Figure 4B*). In contrast to the dose/response profiles (*Figure 3A*), these response/response profiles showed no intermediate peak. This could be because the EGF effect is not visibly included in the response/response profiles, although we cannot exclude the possibility that they were affected by the truncation of the data sets. The uniqueness of TE analysis was again observed in the difference between the response/response profiles and the response/TE profiles. The response/TE profiles were also largely different from the EGF dose/TE profiles (*Figure 3B*). Interestingly, in the late phase, larger causal responses produced only smaller TE in both fw and bw directions, even though the larger causal response induced a larger result response. This could be due to saturation in the response intensities.

## Effects of an MEK inhibitor and a SOS mutation

Previous studies have reported the involvement of negative feedback loops in the EGFR-RAS-MAPK system from ERK to EGFR and SOS (*Corbalan-Garcia et al., 1996*; *Saha et al., 2012*). The detection of significant levels of bw-TE suggests that these feedback loops play a role in the information flow. These feedback loops may be the reactions producing the bw-TE and the multiple TE pathways (*Figure 3E*). To investigate these possibilities further, TE analysis was conducted in cells pretreated with an MEK inhibitor, trametinib. MEK is a protein kinase that mediates activations of RAF and ERK. ERK phosphorylation (activation) caused by MEK activity was examined by changing the concentration of trametinib (*Figure 5A*). In this experiment, 10 ng/ml EGF was used for cell stimulation. The EGF-induced ERK activation was inhibited to the basal level in cells pretreated with 10 nM MEK inhibitor. However, when translocations of SOS and RAF were measured in cells preincubated with 10 nM MEK inhibitor, the SOS and RAF responses were only minimally affected in comparison with the responses after vehicle preincubation (*Figure 5B* and *Appendix 1—figure 5A*). TE was calculated between SOS and RAF responses in these conditions (*Figure 5C* and *Appendix 1—figure 5B*). The MEK inhibitor significantly increased early bw-TE (*Figure 5D*). This result indicates that the feedback from MEK suppresses bw-TE under normal conditions, thereby reducing the control capability of RAF on the SOS response in the early phase of signal transduction.

We also conducted a study on the effects of the R1131K mutation in SOS, which is found in Noonan syndrome patients, with and without the MEK inhibitor. The R1131K mutation had minimal impact on the sensitivity to the MEK inhibitor for ERK phosphorylation (*Figure 5A*) and translocations of SOS and RAF (*Figure 5B* and *Appendix 1—figure 5A*) upon stimulation with EGF. Additionally, it had no significant effect on EGF-induced RAF activation (*Figure 1B*). However, R1131K had a significant impact on the amounts of bw-TE (*Figure 5C and D*). Similar to the MEK inhibitor, R1131K significantly increased the early bw-TE peak. In addition, R1131K specifically decreased the late bw-TE peak. The increase in the early bw-TE peak in cells with R1131K SOS was observed in both the presence and absence of the MEK inhibitor, while the decrease in the late bw-TE peak was observed only in the absence of the MEK inhibitor, meaning a complex relationship between the effects of the MEK inhibitor and R1131K.

## Pathway and regulation of the information flow in the EGFR-RAS-MAPK system

The MEK inhibitor and R1131K SOS mutation affect the feedback from MEK/ERK and the SOS function, respectively. These two factors regulated bw-TE amount in combination (*Figure 6A*). We observed that the regulation system changed according to the time of cell stimulation. In the early phase (<20 min), dysfunction in at least one of the regulation mechanisms, that is the MEK inhibitor or the R1131K mutation, increased the bw-TE compared to the control condition. Whereas, in the late phase (>25 min), only in the absence of the MEK inhibitor, the R1131K mutation decreased the bw-TE.

There could be many different reaction pathways for realizing these types of regulations. Here, we suggest two simple pathways for the early and late phase regulations, respectively (*Figure 6B*). In the early phase, the TE level was normal only when the feedback regulation and the SOS protein were both normal, suggesting a tandem pathway of the two regulations. Based on our prior knowledge

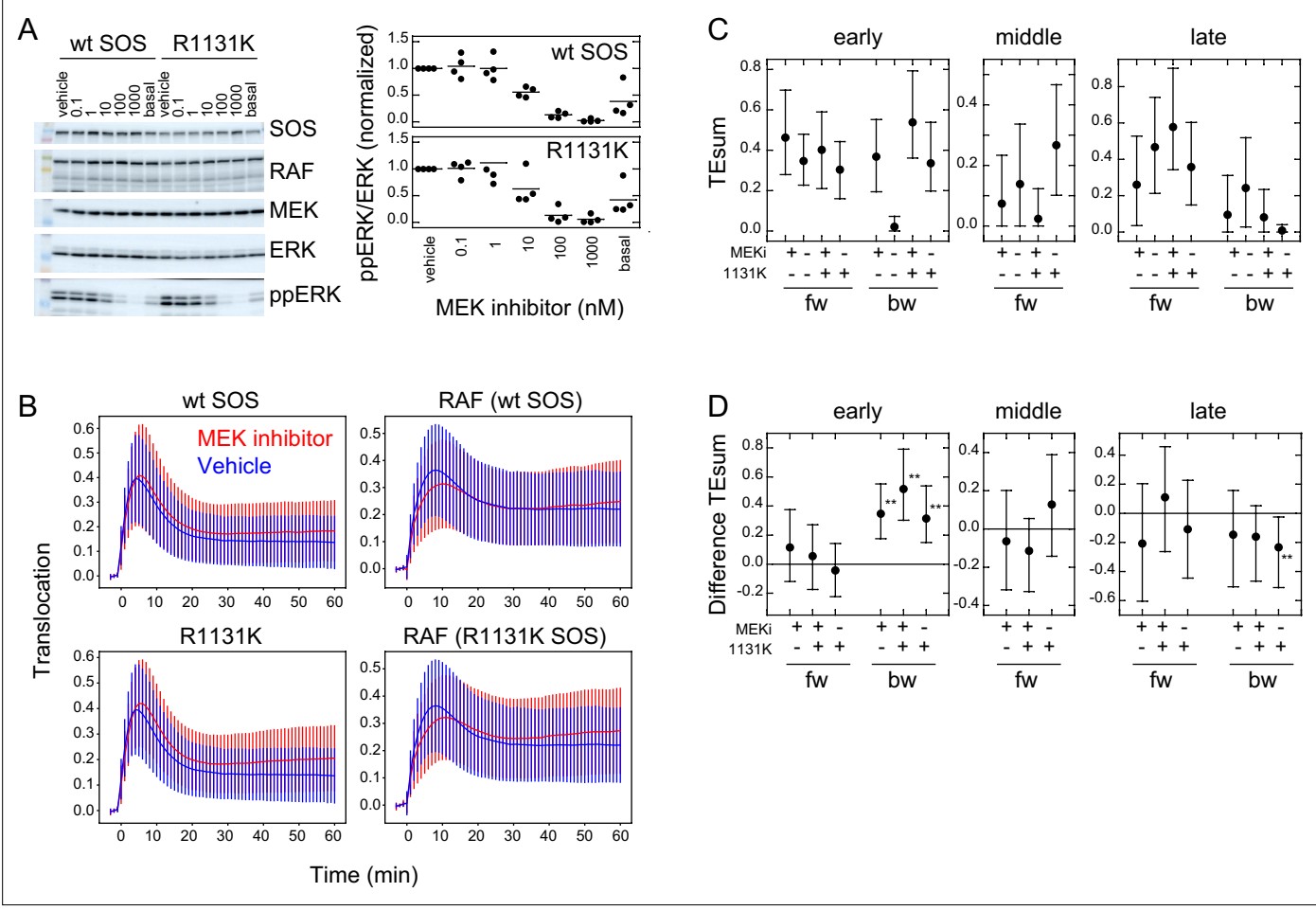

**Figure 5.** Effects of an MEK inhibitor and a Noonan SOS mutation. (**A**) Inhibition of ERK phosphorylation in cells treated with an MEK inhibitor, trametinib. Typical results from western blotting (left) and the inhibitor dose dependency in four independent experiments (right) are shown. Bars indicate the averages. Molecular weight markers indicate 102 (green), 76 (orange), 52 (black), and 38 (blue) kd. (**B**) SOS and RAS responses in cells stimulated with 10 ng/ml EGF. Cells were expressing wt (upper) or R1131K mutant (lower) of SOS and pretreated with 10 nM MEK inhibitor (red) or vehicle solution (DMSO; blue). The numbers of cells were 704 (wt, MEK inhibitor), 744 (wt, vehicle), 832 (R1131K, MEK inhibitor), and 732 (R1131K, vehicle). Averages of cells are plotted with SD. See *Appendix 1—figure 5* for the single-cell time courses. (**C**) TEsum for each peak in the indicated experimental condition. (**D**) Difference between TEsum values under indicated condition and control condition (wt cells with vehicle stimulation). Positive values indicate smaller TEsums in the control condition. **bootstrapping p<0.05. (**C, D**) Averages from 100-time bootstrapping are shown with the 5–95% interval.

The online version of this article includes the following source data for figure 5:

**Source data 1.** Source data for *Figure 5A*.

**Source data 2.** Source data for *Figure 5A*.

about the EGFR-RAS-MAPK system, the MEK/ERK should be upstream of SOS in the feedback path. There are two regulation points between MEK/ERK and the output from SOS affected by the MEK inhibitor and the R1131K mutation, respectively. Since the R1131K mutation affects the SOS function directly, its action point should be downstream. It is also known that the MEK/ERK feedback reduces SOS activity. Therefore, the MEK inhibitor increases the SOS activity. In Noonan syndrome patients, the activity of the RAS-MAPK system is generally enhanced, suggesting that the R1131K mutation also increases the SOS activity. The abnormal regulation at these two points by the MEK inhibitor and the R1131K mutation affects the output from SOS to increase the bw-TE. However, it should be noted that a higher SOS activity does not necessarily result in a higher TE level (as shown in *Figure 4*). The amount of TE is only given in the correlation of the two reactions.

The late-phase regulation may seem odd, but it has a reasonable explanation: In the wt SOS cells, the feedback pathway is mostly disconnected in the late phase, so the MEK inhibitor has little effect

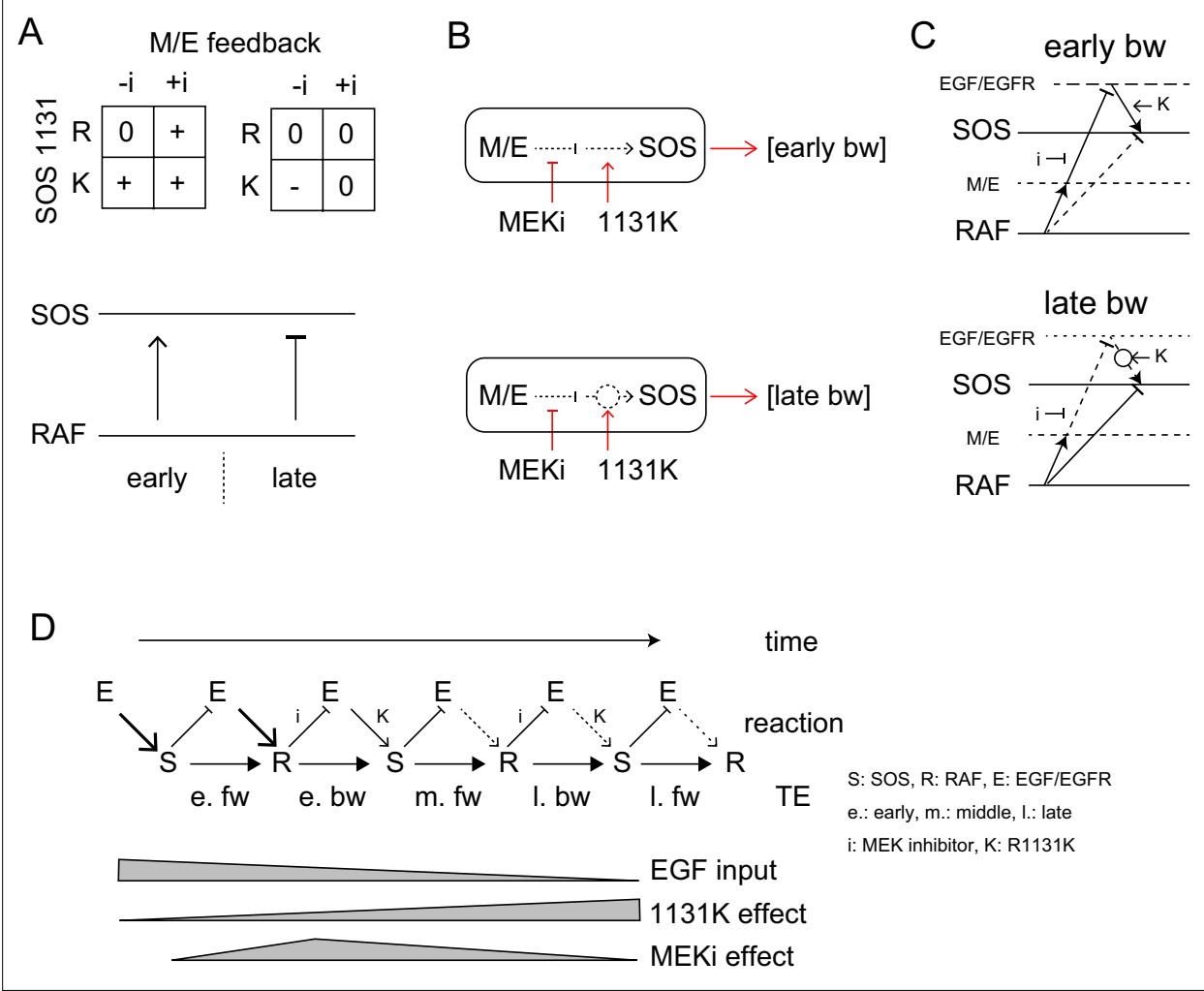

**Figure 6.** TE Regulation by the feedback loop and SOS function. (**A**) 2D matrices showing the effects of the MEK inhibitor and R1131K mutation to the early (left) and late (right) bw-TE. The absence (-i) and presence (+i) of the MEK inhibitor, and the difference in the amino acid residue at SOS 1131 (R or K) caused an increase (+), maintenance (0), or decrease (-) in the TE amount compared to the control condition (-i, **R**). These 2D regulations increased or decreased bw-TE, respectively. (**B**) Simple reaction networks to realize NAND and inversed ANDN gates. The black solid, black dotted, and red arrows indicate TE, reaction, and regulation, respectively. M/E: MEK and ERK. (**C**) Possible reaction networks carrying the bw-TE in the early (upper) and the late (lower) peaks. EGF/EGFR and M/E are the hidden states. These networks are consistent with one of the networks carrying bw-TE suggested in *Figure 3E*. See text for details. (**D**) TE exchange between SOS and RAF involving the EGF/EGFR complex. The role of EGF/EGFR decreases with time after the cell stimulation. The action points of the MEK inhibitor and the R1131K SOS mutation were indicated. See text for details.

on the TE values. The R1131K mutation reconnects the pathway, which restores the effect of the MEK/ERK feedback to decrease the bw-TE amount. However, in the presence of the MEK inhibitor, the feedback loop gets strongly inhibited, and therefore R1131K does not significantly change the TE value. According to this model, the late-phase regulation operates similarly to the early-phase regulation. The only difference between them is the strengths of the connectivity.

The results of the conditional TE analysis to the EGF dose, suggest that there were multiple pathways for the TE. Some pathways include the EGF/EGFR complex, as well as SOS and RAF (*Figure 3E*). The involvement of the EGF/EGFR complex in TE pathways is also supported by the effects of the MEK inhibitor and the R1131K SOS mutation. We constructed a multitrack reaction network that carries bw-TE, including the MEK/ERK feedback loop and activation state of the EGF/EGFR complex (*Figure 6D*). The network for the early phase is consistent with that shown on the left side of *Figure 3Eb*. These pathways are discussed in the 'Discussion' section.

## Discussion

We conducted an assessment of transfer entropy (TE) between SOS and RAF activations in living cells stimulated with EGF. We have observed a significant amount of TE in both directions, from SOS to RAF and from RAF to SOS. There are two possibilities for the generation of bw-TEs: The existence of feedback loops from RAF to SOS and/or the existence of parallel input paths from hidden states to both the SOS and RAF. In both cases, the EGF/EGFR complex is likely to play an important role (*Figure 3Eb*). It is important to note that TE not only signifies the existence of the reaction pathways but also their control capability. The EGFR-RAS-MAPK system is known to involve several feedback loops (*Lake et al., 2016*). For instance, ERK activation downstream of RAF induces EGFR threonine 669 phosphorylation, enhancing EGFR internalization and suppressing EGFR kinase activity (*Li et al., 2008*). ERK also phosphorylates several serine residues in the GRB2-binding domain of SOS (*Rojas et al., 2011*), weakening its interaction with GRB2 and reducing SOS translocation to the plasma membrane upon EGF stimulation (*Corbalan-Garcia et al., 1996*; *Saha et al., 2012*). Furthermore, desensitization of RAF to RAS signal has been reported as a result of multiple serine phosphorylations on RAF by ERK (*Wartmann et al., 1997*). These feedback loops likely carry the bw-TE. Our finding that the MEK inhibitor affected the amount of early bw-TE (*Figure 5*) supports this likelihood. Besides the mentioned negative feedback loops, other factors would complicate the feedback pathway and contribute to the TE amounts, such as multiple membrane binding domains of RAF (*Ghosh et al., 1996*; *Hibino et al., 2011*), RAF phosphorylation on the plasma membrane for activation (*Lavoie and Therrien, 2015*), and the SOS/RAS positive feedback loop (*Margarit et al., 2003*).

After examining the time courses of TE, we found that they had several peaks in both forward and backward directions (*Figure 2*). The second peak from a single causal time must be mediated by hidden states, which created bypasses for TE outside the pathway for the first peak. The conditional TE analysis also suggested that there were multiple tracks of information (*Figure 3E*). One possibility is that some of the multiple tracks pass through EGF/EGFR complex. However, in the fw direction, signaling from SOS to EGFR skipping the MEK/ERK feedback pathway (*Figure 3Ea*, left) has not been proposed as far as we know. A recent study by *Lin et al., 2022* suggests that large complexes consisting of EGFR, GRB2, and SOS are formed by a liquid-liquid phase separation. This may provide a potential pathway for SOS to signal to EGFR. Further research is needed to investigate this possibility. Another possibility is that the EGF/EGFR complex regulates SOS and RAF in parallel. One of the domains in RAF, known as the cysteine-rich domain, has been found to interact with membrane lipids (*Ghosh et al., 1996*). The activation of EGFR can cause a membrane remodeling around EGFR (*Abe et al., 2024*), which could affect the association of RAF with the plasma membrane. The phosphorylation of RAF by kinase molecules on the plasma membrane (*Lavoie and Therrien, 2015*) could also be regulated by EGFR and affect the translocation of RAF. At the present time, the parallel network (*Figure 3Ea* right) is more likely for the fw-TE.

The involvement of the EGF/EGFR complex in the bw-TE pathway was supported by the effects of the MEK inhibitor on TE (*Figures 5 and 6*). A negative feedback from ERK to EGFR has been reported (*Li et al., 2008*). The combination of the MEK inhibitor and R1131K SOS mutation suggested a tandem regulation pathway for the bw-TE (*Figure 6C*). The reaction network containing this regulation pathway (*Figure 6D*, upper) is consistent with the left reaction network in *Figure 3Eb*. A comparable network appears to be utilized for the late bw-TE (*Figure 6D*, lower). However, in the wt SOS cells, the regulation from the EGF/EGFR complex would be weakened in the late phase (as we will discuss later) and would not be detectable in the conditional TE analysis against the EGF dose. The decrease in the middle peak of fw-TE after the MEK inhibitor pretreatment in R1131K cells (*Appendix 1—figure 9*) also suggests that there was a bw-TE pathway via EGF/EGFR in the early phase. There is another feedback pathway exists from ERK to SOS directly (*Corbalan-Garcia et al., 1996*; *Saha et al., 2012*). The MEK inhibitor and R1131K might affect this pathway regardless of the activity of EGF/EGFR. However, considering the time dependency found in the effect of EGF dose on the bw-TE (*Figure 3*), it seems less likely that this regulation was primarily functional.

Next, we consider the time dependency in TE regulation. The extra pathways involving EGF/EGFR were effective in the early phase when the early fw- and bw-TE peaks were detected. A change in TE regulation occurred between 20 and 30 min of EGF stimulation (*Figure 6A*), during which the bw-TE mostly disappeared (*Figure 2*). During this period, the time profiles of SOS and RAF translocation were altered from the initial peak phase to the later plateau phase (*Figure 1D*). The early phase in

the SOS and RAF activation is responsible for transient MEK/ERK activation following EGF signaling (*Marshall, 1995*; *Kao et al., 2001*; *Yoshizawa et al., 2021*). In the late phase, dephosphorylation and/or endocytosis of EGFR reduces signaling from EGFR (*Monast et al., 2012*; *Appendix 1—figure 10*). This decrease in the EGFR signal should be the reason for the insensitivity of the TE levels to the MEK inhibitor in the late phase of wt SOS cells (*Figure 5*). The roles of negative feedback from ERK to EGF/EGFR would be less important after the reduction of EGFR activity.

The information flow was perturbed in Noonan cells carrying the R1131K mutation. The mutation had opposite impacts on the early and late phases in cells without the MEK inhibitor, that is it led to an increase in the early phase and a decrease in the late phase of bw-TE (*Figure 6A*). These seemingly controversial effects can be explained by considering the time dependence in the EGFR signaling intensity, as mentioned above (*Figure 6D, E* and *Appendix 1—figure 10*). Despite suggestions of the involvement of EGF/EGFR complex in the early fw-TE pathway (*Figure 3*), R1131K did not have a significant effect on the early fw-TE (*Figure 5*). Moreover, R1131K did not show any significant impact on the late fw-TE (*Figure 5*). Assuming that the function of R1131K is to sustain the EGFR signaling to SOS by counteracting the negative feedback effect, it will not have any impact before the feedback. In the late phase, R1131K may not be strong enough to boost the weak EGFR signal and alter the quantity of fw-TE (*Figure 6E*). The cell line used in this study expresses SOS2, which has a similar function to SOS1. The function of SOS2 must cause part of the RAF translocation. Our results suggest that R1131K SOS1 has a dominant negative effect on SOS2 and that wt SOS1 at least partially regulates RAF translocation.

The molecular mechanism of onset for most mutants found in Noonan syndrome, including SOS R1131K, is still unknown. After conducting TE analysis, the R1131K mutation is highly possible to affect the EGFR to SOS pathway instead of the SOS to RAF pathway. In other words, the R1131K mutant appears to be defective in receiving input but not in sending output. Furthermore, it was observed that the abnormality caused by R1131K changed with time after cell stimulation. During the early phase, the increase in bw-TE in the R1131K cells competes with fw regulation in the SOS/RAF system. This will prevent the SOS response from precisely following the EGF input. Conversely, during the late phase, R1131K decreases the bw-TE, which relatively strengthens fw regulation. In the late phase, the SOS/RAF feedback loop appears to operate autonomously, outside of the EGF control (*Figure 3*). If inhibition of SOS activation is dominant as the bw regulation, the positive fw and negative bw regulations for the response intensities likely form an adaptation cycle that stabilizes SOS and RAF activation levels (*Ito and Sagawa, 2015*). This type of network could explain the quasi-steady states observed during the late phase (*Figure 1D*). The R1131K mutation disrupts the mutual regulation between SOS and RAF, which can cause the system being out of control. Such abnormality in the information flows caused by R1131K was difficult to detect using conventional biochemical analysis. However, the TE analysis makes it possible to detect bidirectional TE in the steady state without perturbing the system. The phase-dependent effect of R1131K could provide some insight into the medical treatment of Noonan syndrome. The R1131K mutation disrupted feedback regulation. MEK inhibition rescued the abnormality in the late phase but not in the early phase.

This study demonstrated that TE assessment allows for the analysis of intracellular reaction networks from the perspective of reaction control capability. It is important to note that the amount of TE is not necessarily determined by the strength of molecular activation, but rather, by the correlation between the two reactions in single cells. As a result, TE analysis provides a unique perspective of the network that is difficult to obtain with conventional analysis of reaction strengths. In this study, TE analysis successfully predicted the presence of feedback loops and multiple pathways combining SOS and RAF response dynamics without any perturbation to the system and free from any mathematical models. This means that TE analysis is hopeful to be a complement to the conventional ODE analysis. Additionally, TE analysis identifies disorders in reaction regulation caused by a chemical inhibitor and a pathogenic mutation. This shows the applicability of TE analysis to molecular pharmacology and pathology. It is essential to note that the behavior of TE is complex, and gaining more experience with TE analysis will further improve its application in understanding intracellular systems.

## Materials and methods

### Sample preparations

The parental HeLa cell (#RCB0007) was obtained from the RIKEN BioResource Research Center. Methods for the construction of SOS and RAF plasmids and the establishment of SOS1 knockout HeLa cells have been previously described (*Nakamura et al., 2017*). The SOS1 knockout cell was confirmed to be free from mycoplasma infection and the results of its STR polymorphism analysis were identical to those of the original cell line. HeLa cells were cultured in D-MEM supplemented with 10% FBS at 37 °C, 5% $CO_2$. Overnight, cells were transferred onto 25 mm-ϕ coverslips for microscopy or on a 35 mm-ϕ cell culture dish for biochemistry and then transfected with the plasmids using Lipofectamine 3000 (Thermo Fisher). The cells were cultured in MEM containing 1% BSA for one day before being used for the experiments. In some experiments, cells were pretreated with an MEK inhibitor, trametinib (ChemScene) in DMSO (final concentration 0.05%) for 1 hr before EGF stimulation.

### Microscopy

Halo-SOS in cells on a coverslip was conjugated with 300 nM TMR Halo-tag ligand for 1 hr, washed, and SOS and RAF translocations in cells were observed in MEM, 1% BSA, 10 mM HEPES (pH 7.4). An in-house total internal reflection (TIR) fluorescence microscope system, based on IX83 inverted fluorescence microscope (Olympus), was used for imaging. The microscope is equipped with a multicolor laser diode (LDI, Chroma Technology) for fluorophore excitation, a 60 x NA 1.5 oil-immersion objective (UPLAPO60XOHR, Olympus), a dual-color image splitting system (W-VIEW GEMINI-2C, Hamamatsu), and two cMOS cameras (ORCA-Flush4.0, Hamamatsu). Using a 2-dimensional piezo mirror scanner (S-335, PI) in the excitation beam path, this system can alternatively switch the excitation mode between TIR illumination with a circularly rotating laser beam at 40 Hz and epi-illumination, covering a field number of 19 (320 μm in diameter under a 60 x objective). At each time point of the measurement, a set of TIR and epi-illumination images were acquired successively at the same field of view under the excitation laser wavelengths of 470 nm for GFP and 555 nm for tetramethyl-rhodamine (TMR) simultaneously. The exposure time was 125ms and 25ms for each frame in the TIR- and epi-illumination modes, respectively. Image acquisitions were repeated with 1 min intervals. A triple-band dichroic mirror (ZT405/470/555rpc, Chroma) was used in the microscope. In the image splitting optics, the GFP and TMR signals were separated by a dichroic mirror (T550lpxr-UF2, Chroma), and emission filters for GFP (ET510/40 m, Chroma) and TMR (ET575/50 m, Chroma). Just before the 4th time of image acquisition (time 0), cells were stimulated by adding 550 μl of observation medium containing EGF (Pepro Tech) to the 450 μl of the same medium in the observation chamber. All measurements were performed at 25 °C. With 10~13 observation fields, we obtained time-lapse movies of 20~100 cells in a single measurement. The results of six independent measurements under each experimental condition were combined and used for TE calculations, because the number of data from single experiments was too small. Representative responses of cells are shown in *Appendix 1—Videos 1 and 2*. Response curves in the cells expressing wt SOS under 100 ng/ml EGF without MEK inhibitor were obtained under the same conditions as in *Imaizumi et al., 2022*, but from independent measurements.

### Image processing

After the shading and background correction, the outline of single cells in the epi-illumination images was determined using the 'cellpose'" algorithm (https://github.com/mouseland/cellpose; *Pachitariu and Stringer, 2022*; *Stringer, 2025*) and the average intensities per pixel in the single cell areas were measured in each video frame. Data from cells with the lower 20% of signal intensities at time 0 were removed, because it was difficult to separate from background noise. From the single-cell time courses of the average intensity, bleed-through between the GFP and TMR fluorescence was subtracted, drift in the basal intensity with time was removed, then the TIR signal was normalized to the epi signal. The difference in the normalized TIR signal to the 3-frame average before stimulation was plotted as the single-cell response. Image processing was performed using Image J (NIH). The response dynamics data used in this study have been deposited to https://github.com/YasushiSako/transfer_entropy_2/.

## Transfer entropy calculations

Transfer entropy (TE) values from SOS to RAF and RAF to SOS were calculated along the time courses of the cell responses. The source code for the calculations has been deposited to https://github.com/YasushiSako/transfer_entropy_2/, which is a modification of *Imaizumi et al., 2022*, TE_by_Gaussian_Approximation.ipynb, https://github.com/kabashiy/transfer_entropy/. The method used in these programs is precisely described by *Imaizumi et al., 2022*.

In brief, when assessing the uncertainty of the stochastic time sequence of $X$ at time $t$ ($X_t$) by the statistical entropy $H(X_t)$, the amount of the uncertainty reduction after obtaining the past sequence of $X$ at the reference time point Tr, $X_{Tr}$ is $I\left(X_t; X_{Tr}\right) = H\left(X_t\right) - H\left(X_t | X_{Tr}\right)$. Here, $H\left(X_t | X_{Tr}\right)$ is the conditional entropy, and $I$ is mutual information. The TE from $Y$ to $X$ is defined as $TE_{Y \to X}^{Tc,Tr}\left(t\right) = I\left(X_t; X_{Tr}, Y_{Tc}\right) - I\left(X_t; X_{Tr}\right)$, which represents the reduction of uncertainty about $X_t$, given the additional information from $Y$ at the causal time point Tc, $Y_{Tc}$, beyond what is already provided by $X_{Tr}$. The calculation of TE from real experimental data is usually difficult due to insufficient data numbers to accurately estimate the multi-dimensional distribution of the generating function. Even if the data number is sufficient, a large amount of computational power is required. However, in cases where the joint distribution of **X** and **Y** is multivariate Gaussian, TE can be analytically evaluated using covariance matrices for the Gaussian model as

$$TE_{Y \to X}^{Tc,Tr}\left(t\right) = \frac{1}{2} ln \left( \frac{\sum \left(X_t | X_{Tr}\right)}{\sum \left(X_t \vee X_{Tr} \oplus Y_{Tc}\right)} \right).$$

Here, $\sum\left(X\right)$ and $\sum\left(X, Y\right)$ denote the covariance matrix of $X$ and the cross-covariance matrix of $X$ and $Y$, respectively, and $\sum(X_t | X_{Tr}) = \sum(X_t) - \sum(X_t, X_{Tr}) \sum(x_{Tr})^{-1} \sum(X_t, X_{Tr})^{\mathrm{T}}$. $X \oplus Y$ refers to the concatenation of the matrices $X$ and $Y$. Procedure of TE calculation from the measured data set is shown in *Appendix 1—figure 1*. Gaussian approximation is the first approximation but the only practical way to realize TE calculation from noisy real data sets. Even in non-Gaussian cases, it is known that non-zero TE values under the Gaussian approximation mean non-zero values of the true TE (*Marinazzo et al., 2008*).

To determine the significance threshold, we first generate a sample set by random sampling and calculate the TE with the null hypothesis, TE = 0. This means that we use a setting in which all elements of the cross-covariance between $X_t \oplus X_{Tr}$ and , $Y_{Tc}$ are vanished. Then, we obtain the distribution of TE values by resampling the data 1000 times. From this distribution, we adopt the upper 1% percentile point as the significant threshold. Finally, if the TE value exceeds this threshold, we conclude that the TE value is statistically significant. This procedure was repeated 100-times for the randomly sampled data, allowing duplication (bootstrapping), to obtain the average and 5–95% range. The number of sampled data was the same as the number of original data number unless otherwise noted.

## Western blot analysis

Cells under the described conditions were solubilized and separated on 10% polyacrylamide gels, and transferred to polyvinylidene difluoride membranes (BD Biosciences). The membranes were incubated with an anti-SOS1 (#5890, Cell Signaling Technology), anti-RAF (#610151, BD Bioscience), anti-RAF pS338 (05–538, Millipore), anti-EGFR pY1068 (3777 S, Cell Signaling Technology), or anti-β-action (#5441, Sigma), and a secondary antibody conjugated with horseradish peroxidase (#7074, #7076, Cell Signaling). Antibody bindings were visualized with ECL Prime Western Blotting Detection Reagent kit (Cytiva) and imaged by using ImageQuant LAS500 (GE Healthcare).

## Numerical simulations

Stochastic simulations of reaction cascade models were performed using the program described in *Shindo et al., 2018*, where Langevin ODE models were solved using the 4th-order Runge-Kutta method. The Supplementary Text provides details of these models and the results of TE analyses of the models. The purpose of analytical modeling is to assess the behavior of TE conceptually, not to reproduce the real intracellular reaction network.

## Legend for source data

'TE_by_Gaussian_Approximation_2.ipynb' is the Python source code for the TE calculation. Data files for protein translocation dynamics are contained in the two folders 'EGF dose dynamics data'

and 'MEKi 1131K dynamics data' in csv format. Each data file can be read directly by the Python code.

## Acknowledgements

The authors thank Mutsumi Nakanishi and Hiromi Sato for technical assistance, Sinsuke Uda at Kyushu University, Atsushi Hatano at Nigata University, and Tomoyuki Obuchi at Kyoto University for helpful discussions and suggestions. The SOS knockout Hela cell was established by Mitsuhiro Abe at RIKEN. YS and YK were funded by JST CREST (JPMJCR1912). YS and YK were funded by Grants-in-Aid for Scientific Research, MEXT (19H05647) and (22H05117), respectively.

## Additional information

### Funding

| Funder | Grant reference number | Author |
|---|---|---|
| Collaborative Research in Engineering, Science and Technology Centre | JPMJCR1912 | Yoshiyuki Kabashima Yasushi Sako |
| Ministry of Education, Culture, Sports, Science and Technology | 19H05647 | Yasushi Sako |
| Ministry of Education, Culture, Sports, Science and Technology | 22H05117 | Yoshiyuki Kabashima |

The funders had no role in study design, data collection and interpretation, or the decision to submit the work for publication.

### Author contributions

Nobuhisa Umeki, Data curation, Formal analysis, Investigation, Writing – original draft, Writing – review and editing; Yoshiyuki Kabashima, Software, Methodology, Writing – original draft, Writing – review and editing; Yasushi Sako, Conceptualization, Resources, Software, Formal analysis, Supervision, Funding acquisition, Methodology, Writing – original draft, Writing – review and editing

### Author ORCIDs

Yasushi Sako https://orcid.org/0000-0002-5707-5455

### Decision letter and Author response

Decision letter https://doi.org/10.7554/eLife.104432.sa1
Author response https://doi.org/10.7554/eLife.104432.sa2

## Additional files

### Supplementary files

MDAR checklist

### Data availability

The response dynamics data used in this study have been deposited to https://github.com/YasushiSako/transfer_entropy_2/. The source code for the calculations has been deposited to https://github.com/YasushiSako/transfer_entropy_2/ (copy archived at *Sako, 2025*), which is a modification of https://github.com/kabashiy/transfer_entropy/ (*kabashiy, 2022*).

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

# Appendix 1

## TE and conditional TE

Statistical entropy:

The uncertainty of the system variable *x* in a multistate system X can be described by the statistical entropy H(X), which is calculated from the probability distribution of *x*, p(*x*) as follows:

$$H\left(X\right) = \sum_x -p\left(x\right) log p\left(x\right).$$

Since uncertainty disappears once a measurement specifies the value of *x*, statistical entropy refers to the amount of uncertainty reduction, that is the amount of information obtained by the measurement.

Mutual information:

When there is a relationship (correlation) between two coexisting systems, X, Y (with system values *x* and *y*), identifying the state of one reduces the uncertainty about the state of the other. This reduction is called mutual information, MI(X;Y). The mutual entropy can be calculated from p(*x*), p(*y*), and the joint distribution of *x* and *y*, p(*x*,*y*):

$$MI = H\left(X\right) - H\left(X \vee Y\right) = H\left(X\right) + H\left(Y\right) - H\left(X, Y\right) = \sum_x \sum_y p\left(x, y\right) log \frac{p\left(x, y\right)}{P\left(x\right) p\left(y\right)}.$$

H(X|Y) is the entropy of X conditioned on Y (see later). MI is a symmetric value with respect to X and Y, and means the amount of information that X (or Y) has with respect to Y (or X).

Transfer entropy:

Assuming that X and Y are time-varying, we denote their values at time *t* as *x*(*t*) and *y*(*t*) and consider estimating *y*(*t*) from *y* at a past time Tr, *y*(Tr). Tr may not be a single time point but may have a certain time width. As above, the estimation accuracy can be described as the mutual information MI($Y_t$; $Y_{Tr}$). For this estimation, we need a simultaneous distribution of *y*(Tr) and *y*(*t*).

If we have a tripartite simultaneous distribution of *x*(Tc), *y*(Tr), and *y*(*t*), we can calculate the mutual information, MI($Y_t$; $X_{Tc}$, $Y_{Tr}$), using the distribution of *y*(*t*) and the joint distribution of *x*(Tc) and *y*(Tr). Here we estimate *y*(*t*) using information from *x*(Tc) and *y*(Tr) simultaneously. The difference between MI($Y_t$; $X_{Tc}$, $Y_{Tr}$) and MI($Y_t$; $Y_{Tr}$) is called the transfer entropy, TE, from X to Y.

$$TE_{X \rightarrow Y} = MI\left(Y_t; X_{Tc}, Y_{Tr}\right) - MI\left(Y_t; Y_{Tr}\right) = MI\left(Y_t; X_{Tc} \vee Y_{Tr}\right)$$

Due to the non-negativity of the mutual information, TE is non-negative. Significantly large $TE_{X \rightarrow Y}$ means that the accuracy of the estimation of *y*(*t*) has increased by considering the information in *x*(Tc). When Tc, Tr <*t*, a significantly large $TE_{X \rightarrow Y}$ indicates that X(Tc) has some sort of causality to Y(*t*). TE is asymmetric for X and Y as shown in the definition above. $TE_{X \rightarrow Y}$ is a quantity that expresses how much X(Tc) controls Y(*t*), independently of Y(Tr) (control capability). Note, however, that TE may appear due to spurious correlations. Even in this case, the TE still provides some insight into the system.

Conditional TE:

If another system, Z is measured at the same time as X and Y, we can consider the distribution of *x* (or *y*) and the joint distribution of *x* and *y* under a given value of Z. These are the conditional probability distributions given *z*, p(*x*|*z*) and p(*x*,*y*|*z*). The mutual information calculated from the conditional probability distributions is called conditional mutual information.

$$MI\left(X; Y \vee Z\right) = \sum_z \sum_x \sum_y p\left(z\right) p\left(x, y \vee z\right) log \frac{p\left(x, y \vee z\right)}{P\left(x \vee z\right) p\left(y \vee z\right)}$$

We can calculate conditional TE from conditional mutual information:

$$TE_{X \rightarrow Y \vee Z} = MI\left(Y_t; X_{Tc}, Y_{Tr} \vee Z\right) - MI\left(Y_t; Y_{Tr} \vee Z\right)$$

By the definition of MI:

$$TE_{X \to Y} = H\left(Y_t | Y_{Tr}\right) - H\left(Y_t \vee X_{Tc}, Y_{Tr}\right),$$

and

$$TE_{X \to Y \vee z} = H\left(Y_t | Y_{Tr}, Z\right) - H\left(Y_t \vee X_{Tc}, Y_{Tr}, Z\right)$$

We defined the difference TE (ΔTE) as:

$$TE_{X \to Y \vee z} - TE_{X \to Y} = \{H(Y_t|Y_{Tr}, Z) - H(Y_t|Y_{Tr})\} + \{H(Y_t|X_{Tc}, Y_{Tr}) - H(Y_t \vee X_{Tc}, Y_{Tr}, Z)\}.$$

In general, due to the non-negativity of the information carried by Z, the uncertainty in $Y_t$ does not change or decrease after conditioning on Z, i.e, $H\left(Y_t|W, Z\right) \leq H\left(Y_t \vee W\right)$. Here, W is any kind of distribution. Therefore, the value in the first parenthesis of ΔTE is non-positive and the value in the second parenthesis is non-negative. A negative ΔTE means that at least the first value is negative, that is Z correlates with $Y_t$ independently of $Y_{Tr}$. While a positive ΔTE means that at least the second value is positive, that is Z correlates with $Y_t$ independently of $X_{Tc}$ and $Y_{Tr}$. A tandem reaction path, $Z \to X_{Tc} \to Y_t$ does not satisfy the latter condition, so its ΔTE cannot be positive. But ΔTE can be negative because Z has a control path to $Y_t$ that is not mediated by $Y_{Tr}$. Among the simple networks of X, Y, and Z, the pathways where Z controls $Y_t$ independently of $X_{Tr}$ and $Y_{Tc}$ to have the potential to produce positive ΔTE in the forward direction are $X_{Tc} \leftarrow Z \to Y_t$ and $X_{Tc} \to Z \to Y_t$ (**Figure 3E**).

## TE analysis in biology

Here are some examples of TE analysis applied to biology. TE analysis can be applied to any multidimensional data, including the results of molecular dynamics simulations. *Kamberaj and van der Vaart, 2009* used TE analysis to extract the causality of correlated motions within single molecules from molecular dynamics simulations of a DNA-binding protein molecule, ETS-1 transcription factor. They found that the direction of information flow changes upon DNA binding. This work revealed the mechanism of allosteric protein dynamics.

*Wibral et al., 2011* analyzed TE in MEG data from a human auditory short-term memory experiment to estimate the network topology of brain modules. The topology was task type specific. *Stetter et al., 2012* developed a method to infer the neural network connectivity from the TE between the series of single-cell calcium responses in the network. This method outperformed previous methods in simulations, and was applied to the experimental data of cultured neuronal cell populations. *Ursino et al., 2020* developed another TE-based method for estimating network structure and connection strengths that is applicable to complicated 2–4 neuron networks. They reported that in the liner regime of the cell response, there was a fairly good correlation between TE and synaptic strength when the data lengths were sufficient. These are examples of the application of TE to neural circuit analysis.

*Pahle et al., 2008* performed stochastic simulations of the association between $Ca^{2+}$ binding to a protein and developed a framework for investigating the relationship between the patterns of $Ca^{2+}$ dynamics and the ability to transfer information from the $Ca^{2+}$ dynamics to the $Ca^{2+}$ binding level of the protein. *Ito and Sagawa, 2015* showed theoretically that the robustness of an information processing system to environmental changes can be quantitatively evaluated by TE. They analyzed the experimentally obtained adaptive behavior of *E. coli* chemotaxis signal transduction, and found that chemotaxis signal transduction is efficient as an information transmission device despite being highly dissipative as a thermodynamic engine.

## Stochastic ODE simulations

### Models and simulation

We consider reaction networks between X, Y, and Z (**Appendix 1—figure 11A**):

$$\frac{dY}{dt} = \left(a_{12} X\left(t\right) - a_{32} Z + a_2\right)\left(T_Y - Y\right) - b_2 Y + \xi_Y,$$

$$\frac{dZ}{dt} = \left(a_{13} X\left(t\right) - a_{23} Y + a_3\right)\left(T_Z - Z\right) - b_3 Z + \xi_Z.$$

Here, X is the input to Y and Z set to be a single-peaked two-phase function as follows:

$$\frac{dX_0}{dt} = -a_0 X_0 + \xi_{X0},$$

$$\frac{dX_1}{dt} = \left(a_0 X_0 - a_{21} Y - a_{31} Z\right) - b_{01} X_1 + \xi_{X1},$$

$$\frac{dX_2}{dt} = \left(a_0 X_0 - a_{21} Y - a_{31} Z\right) - b_{02} X_2 + \xi_{X2},$$

$$X\left(t\right) = X_1\left(t\right) + X_2\left(t\right)/5$$

For numerical simulations of the above model, $a_0$, $b_{01}$, and $b_{02}$ were set to 1, 0.1, and 0.005, respectively, to mimic the phosphorylation dynamics of EGFR in real cells qualitatively (*Appendix 1—figure 10*). $a_2 = a_3 = 0.01, b_2 = 0.5, b_3 = 0.1$, and $T_Y = T_Z = 10$. $\xi_i$ is a random parameter obeying a Gaussian distribution with the average = 0 and the standard deviation = $D\sqrt{F\left(t\right)}$ for each *dF/dt*, and *D* is a variable parameter controlling the noise level. *D*=0.01 was used for all simulations. Other parameter values are listed in *Appendix 1—figure 11A*. By setting some parameter values to zero, we changed the network topology of the models. All of the models we examined here contain an input path from X to Y. $X_i$ shown in *Appendix 1—figure 11A* is the initial value of $X_0$ and controls the input strength. These parameter values were chosen arbitrarily to obtain transient responses of Y and Z with unimodal intensity distributions in the range of $X_i$.

For each model, four simulations of reaction time courses were performed by varying $X_i$ from 0.02 to 0.35 in increments of 0.05 (*Appendix 1—figure 11*). In the simulations, Y and Z were first equilibrated in the absence of input (X(t)=0) for *t* = –50–0 (5,000 steps in the simulations) from appropriate initial values, and then Y(0) and Z(0) were used for the initial values of the simulation with input X(*t*>0). The trajectories of Y and Z were calculated 500 times for each $X_i$ during *t* = –3~60.

## TE and conditional TE

First, TE time courses between Y and Z were calculated for each model using all data sets varying $X_i$ as the averages of 100 bootstrap runs (*Appendix 1—figure 11B*). In the tandem reaction model (X→Y→Z), significant amounts of TE were observed in the forward (Y→Z) direction, as expected. Amounts of bw-TE (Z→Y) were small and should be a statistical fluctuation. Adding a feedback loop from Z to Y or X (feedback models: X→Y↔Z and X→Y→Z→ X) increased bw-TE amounts. The size of TE in the parallel input model $\left(Y \leftarrow X \rightarrow Z\right)$ and depended on the reaction rate constants. When the reaction rate constant from X to Y ($a_{12}$) was larger than that from X to Z ($a_{13}$), fw-TE was dominant, but when $a_{12}$ was smaller than $a_{13}$, bw-TE became dominant. These results are not surprising given the spurious causality between Y and Z in this model. The X-mediated model $\left(Y \leftrightarrow X \rightarrow Z\right)$ does not contain feedback loops from Z to Y, and its bw-TE level was low. Thus, simulations suggest that detection of a significant amount of bw-TE requires a feedback loop from Z or parallel inputs to Y and Z from X.

Next, the difference between TEs (ΔTE) conditioned and non-conditioned to $X_i$ was examined (*Appendix 1—figure 11C*). Conditional TE is the average of the TEs for each $X_i$ value, while non-conditional TE is the TE for mixed data for all $X_i$ values. We performed 100 times bootstrapping for the non-conditional and conditional TEs and calculated ΔTE = conditional TE – non-conditional TE for each bootstrap run. ΔTE was calculated only between the statistically significant TE values. Significant positive ΔTE was observed only in the forward direction of the Z to X feedback model (X→Y→Z→X), where input X mediates the information path from Y to Z as expected (Y→Z→X→Y→Z). Negative significant ΔTE has been observed in some conditions where X is connected to $Z_t$ (in fw direction) or $Y_t$ (in bw direction) with skipping $Z_{Tr}$ or $Y_{Tr}$, respectively. This is also in line with expectations. However, it is not easy to explain what determines the sign and amount of ΔTE.

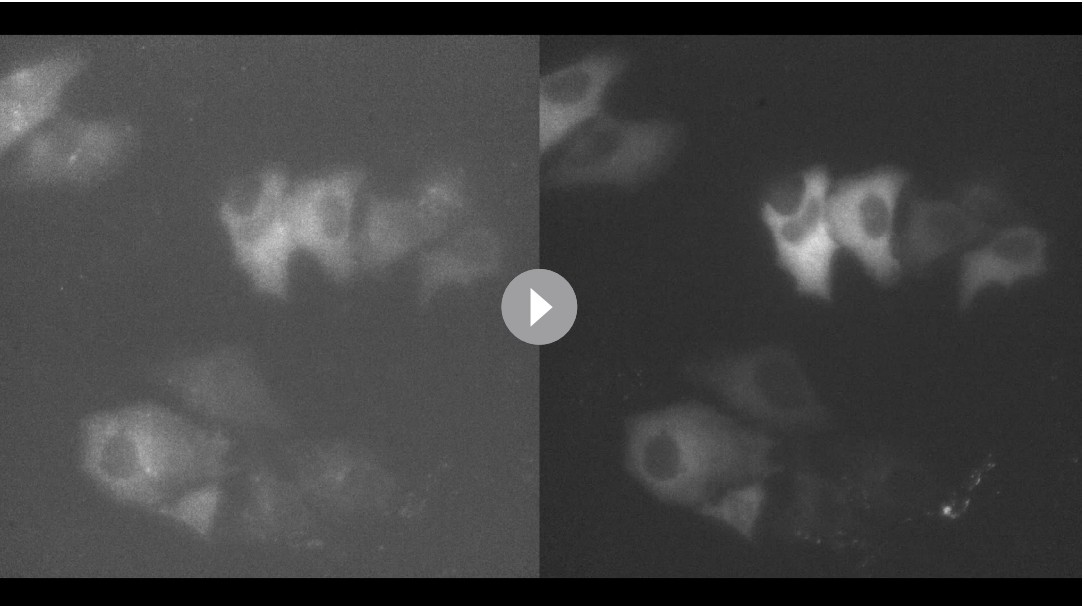

**Appendix 1—video 1.** SOS and RAF in cells stimulated with EGF.
https://elifesciences.org/articles/104432/figures#video1

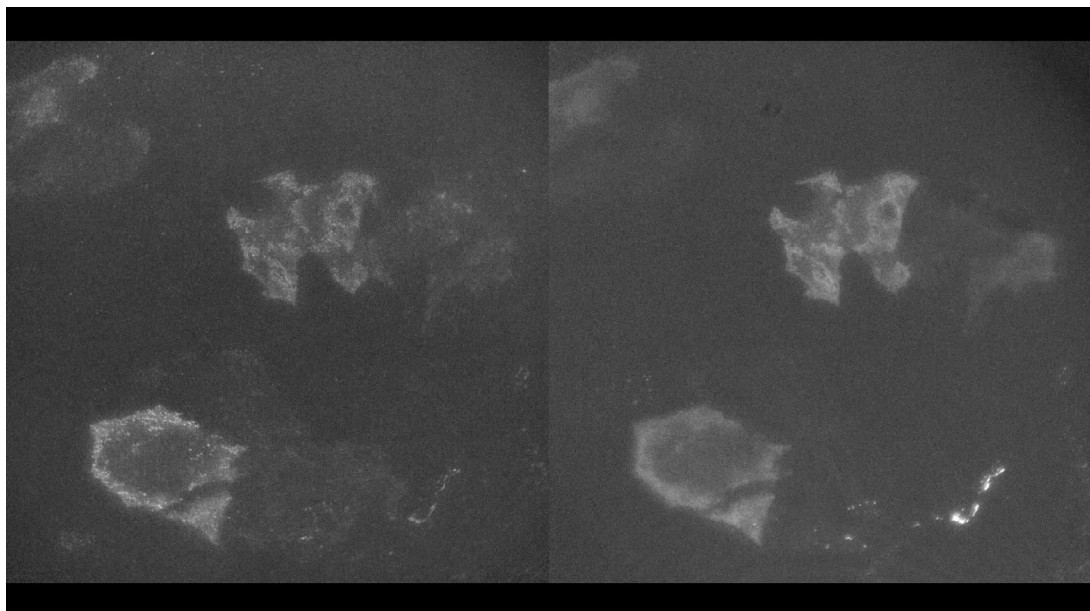

**Appendix 1—video 2.** SOS and RAF translocation to the basal cell membrane in cells stimulated with EGF. Time-lapse movies with a 1-min interval for 63 min. These two movies were acquired in the same field of view under epi (Movie S1) or total internal reflection (Movie S2) illumination. The left and right movies show signals from Halo (TMR)-SOS and GFP-RAF, respectively. Just before the 4th frame, 100 ng/ml (final concentration) of EGF was added to the observation medium. The field of view is 222 x 222 μm$^2$.
https://elifesciences.org/articles/104432/figures#video2

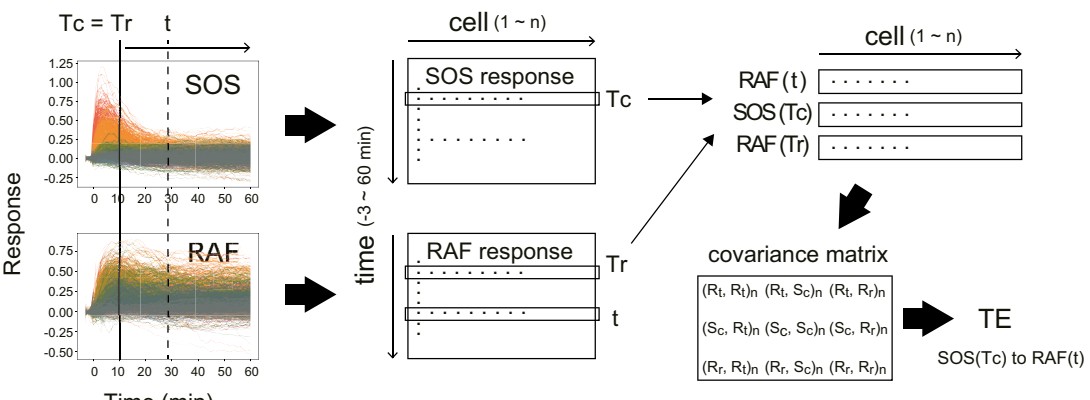

**Appendix 1—figure 1.** Calculation of transfer entropy. The procedure for the calculation of TE from SOS(Tc) to RAF(*t*) with reference to RAF(Tr) is shown. (Here, Tr = Tc.) From the single-cell dynamics of SOS and RAF translocations (left), the single-cell response distributions SOS(Tc), RAF(Tr), and RAF(*t*) were extracted (middle to upper right; cell number = n). Then, the covariance matrix between the extracted distributions (bottom right) was caluscted, from which the TE was calculated according to the formula shown in the 'Materials and methods' section. R: RAF, S: SOS, c: TC, r: Tr. ($\mathbf{X}_i$, $\mathbf{Y}_j$): covariance beween $\mathbf{X}(i)$ and $\mathbf{Y}(j)$. ($\mathbf{X}_i$, $\mathbf{X}_i$) is the variance of $\mathbf{X}(i)$.

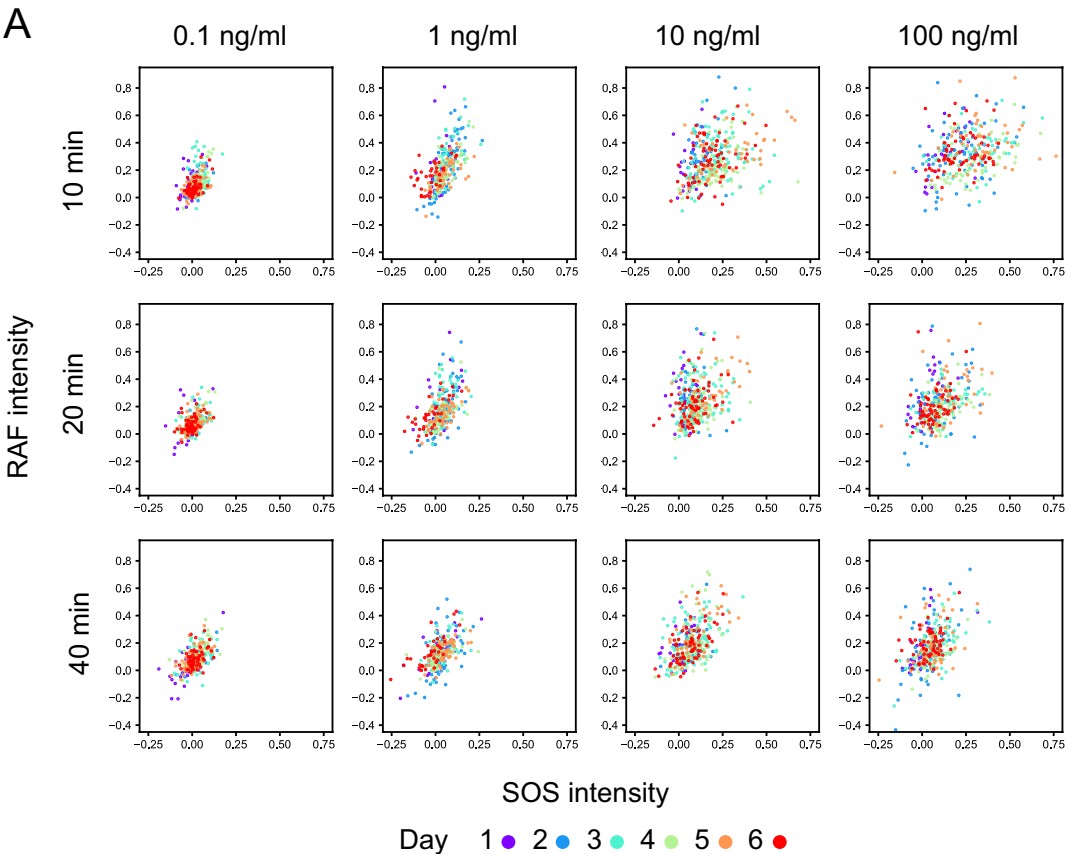

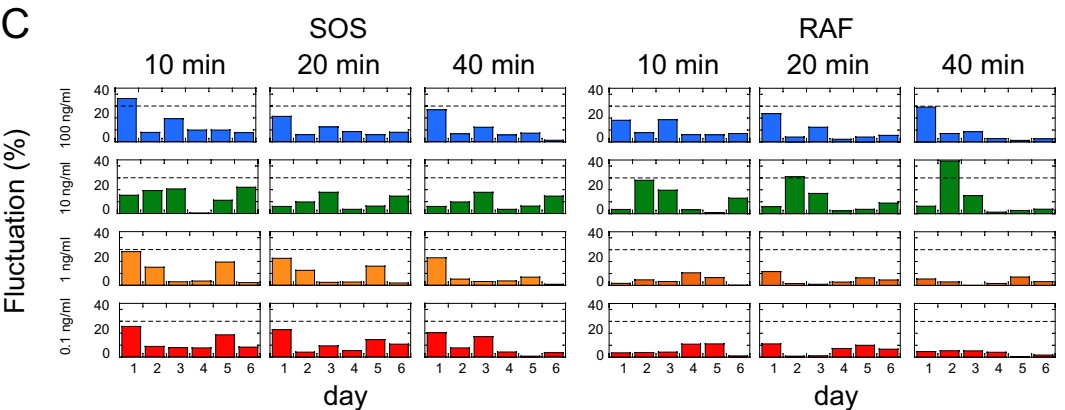

**Appendix 1—figure 2.** Measurement reproducibility. (**A**) Single-cell response intensities of SOS and RAF at three different time points after EGF stimulation, with EGF dose indicated. Different colors represent measurements obtained on different days. Responses on each day overlap with a small bias. (**B**) Numbers of cells shown in A. *Appendix 1—figure 2 continued on next page*

*Appendix 1—figure 2 continued*
(**C**) Inter-day variation in response mean. Variation was defined as the absolute value of the difference between the daily mean and the overall mean divided by the 5–95% range of the total single-cell response. The dashed line indicates a 30% variation.

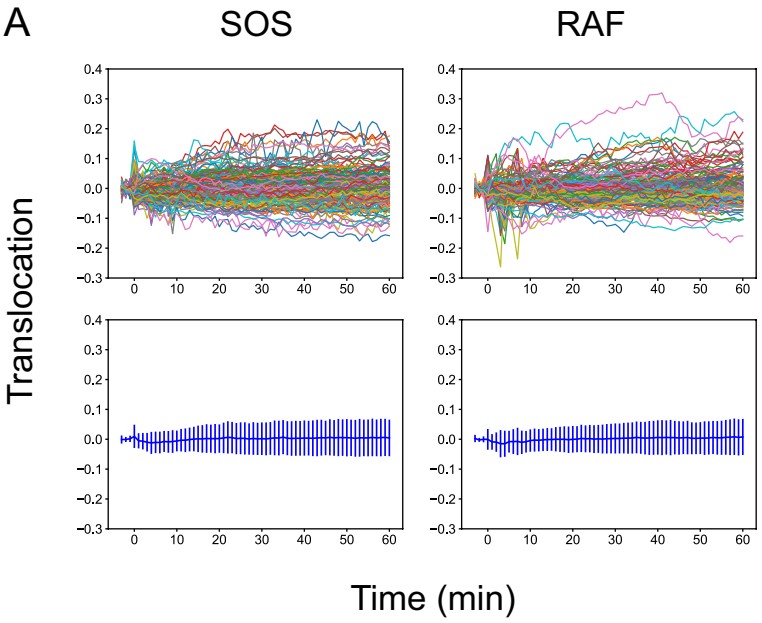

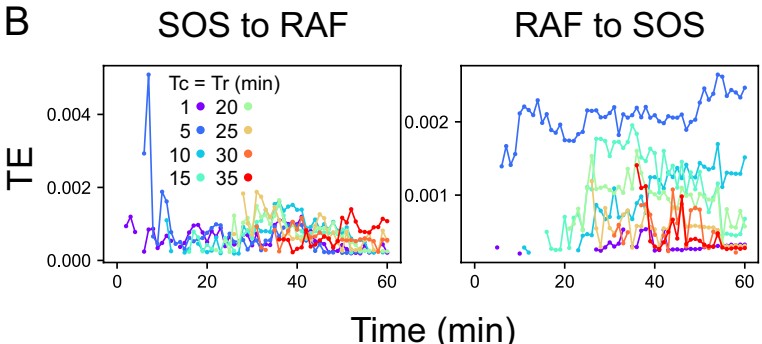

**Appendix 1—figure 3.** Reactions and TEs under vehicle stimulation. (**A**) Time courses of SOS and RAF responses after vehicle stimulation. These are the same plots shown as the black lines in *Figure 1C*. The upper panels show single-cell trajectories (n=150). The bottom panels show the mean with SD. No significant responses were observed. (**B**) Time courses of TE as the average of 100 times bootstrapping. The TE values are one order of magnitude lower than those observed after EGF stimulation (*Figure 2*). These values could be noise caused by the positive bias of TE or true information transfer between basal responses in quiescent cells. Since there were no obvious temporal structures, it is less likely that the TEs were caused by the effect of the addition of vehicle solution.

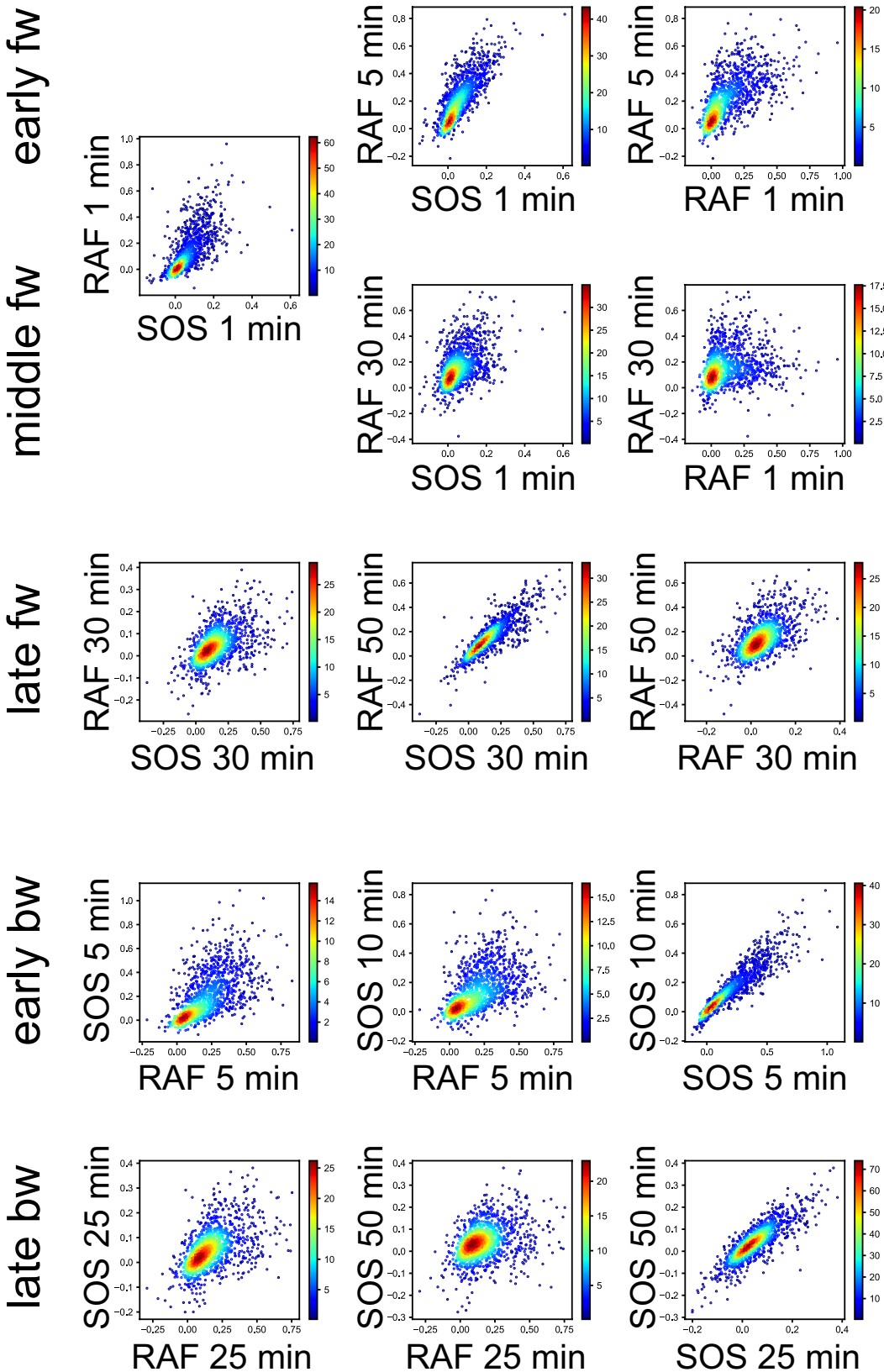

**Appendix 1—figure 4.** 3D response intensity distributions. Single-cell intensity distributions at the representative Tc, Tr, and *t* in TE time courses are shown as the 2D-projections. Colors indicate the densities of single-cell data points in a normalized unit.

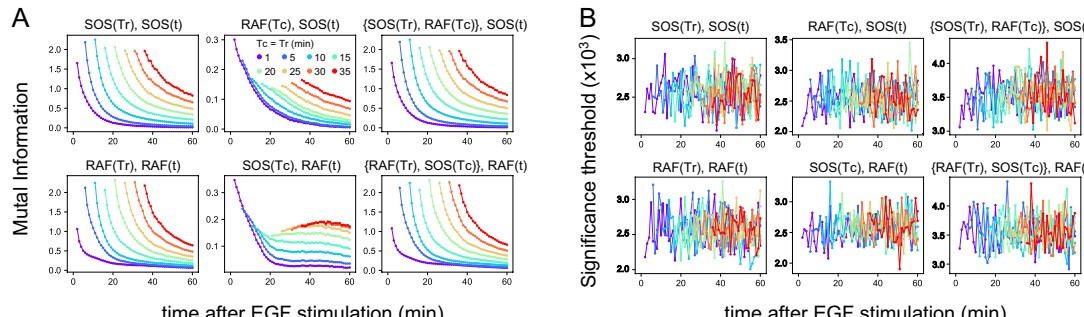

**Appendix 1—figure 5.** Mutual information. (**A**) Time courses of the mutual information between the reaction intensities were calculated using the total 1317 data sets (*Figure 1D*). The mutual information (MI) contents, the values of the two terms of TE (*Figure 2A*) at the times Tc, Tr, and *t* used for TE calculation (*Figure 2B*) are shown. Values higher than the upper 1% values in the 1000 times bootstrapping calculation under the null hypothesis were plotted. (In this criterion, all values were statistically significant in these calculations.) (**B**) The significance threshold for the calculation in A, which is an indication of the error values in A. The thresholds for MI(X(Tr); X(t)) (~0.001) were significantly smaller than the values of TE(Y(Tc) to X(t)) (~0.01), supporting that the results of TE calculations were meaningful and sufficiently larger than the calculation errors in the subtractions between the two terms of TE.

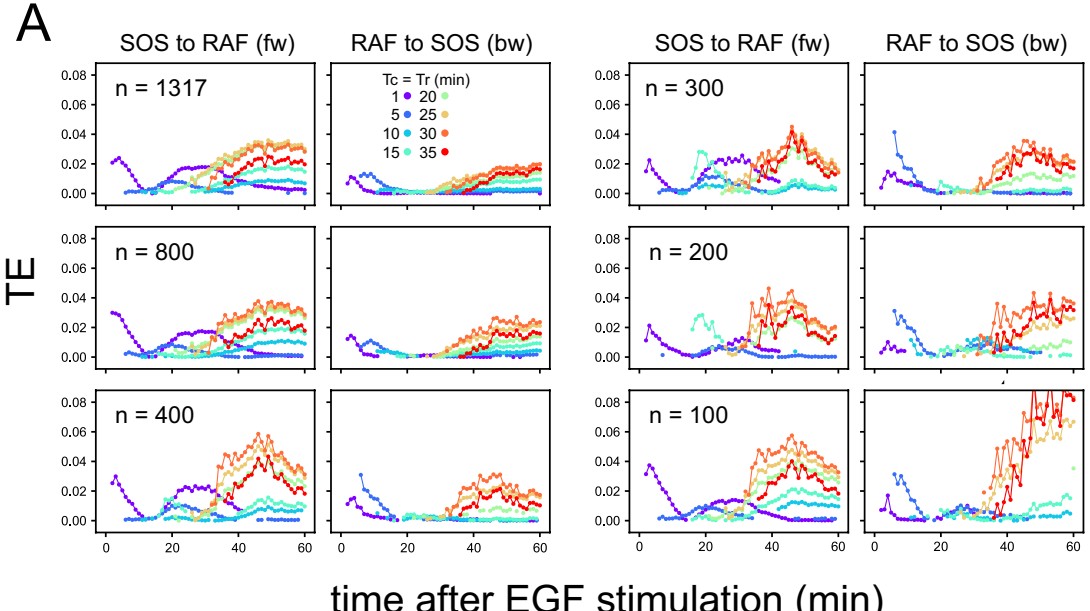

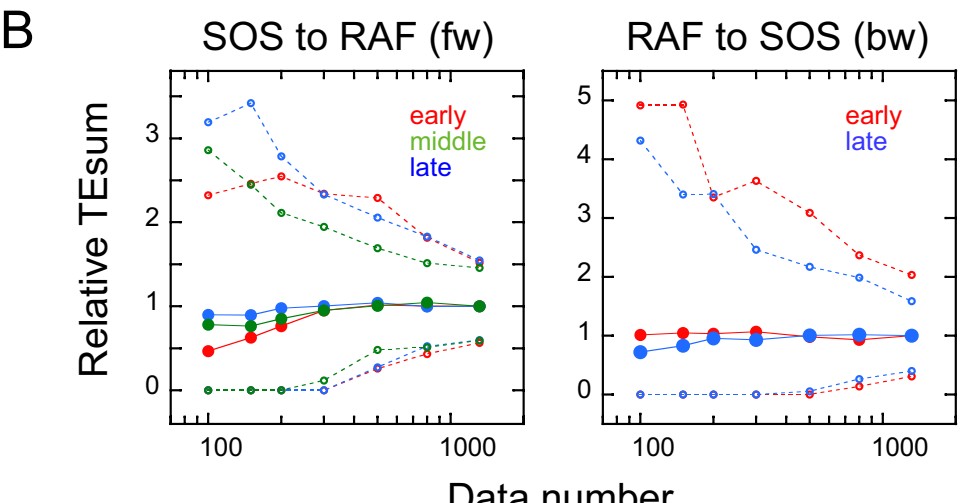

**Appendix 1—figure 6.** Effects of the data number on transfer entropy detection. Since the TE takes only non-negative values and the TE time courses are noisy, we checked if the limited data number caused a false detection of significant TE, reducing the number of data sequences used for calculation. Here, 100–800 data sets were randomly selected from the total 1317 data sets (*Figure 1D*). (**A**) Examples of TE time courses calculated using different numbers of data sets (**n**). As the number of data sets decreases, the number of time points where a statistically significant level of TE was detected also decreases. (These plots only show the statistically significant values.) This result means that TE was not detected falsely due to the limited number of data. (**B**) TEsum values for 100 times of bootstrapping as a function of the data number. Values are normalized to those for 1317 data sets. The average (solid line) and 5 and 95% values (dotted lines) are shown. Even though the decrease in the data number caused the increase in the uncertainty (5–95% range), the averages were still within the 83–104% range of the TE for the total of 1317 data sets (excluding those for the early fw peak), when n≥150. For the early fw peak, the bootstrap averages were within the 63–107% range.

A

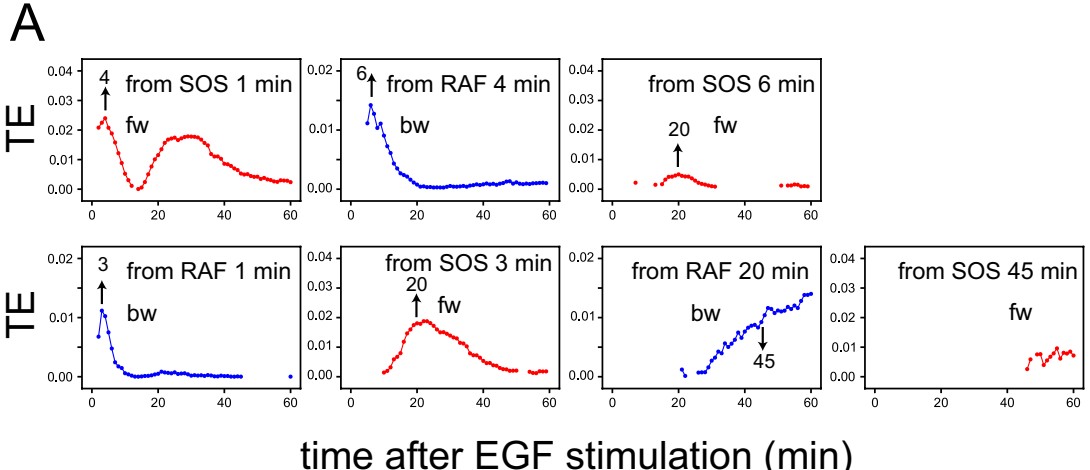

time after EGF stimulation (min)

B

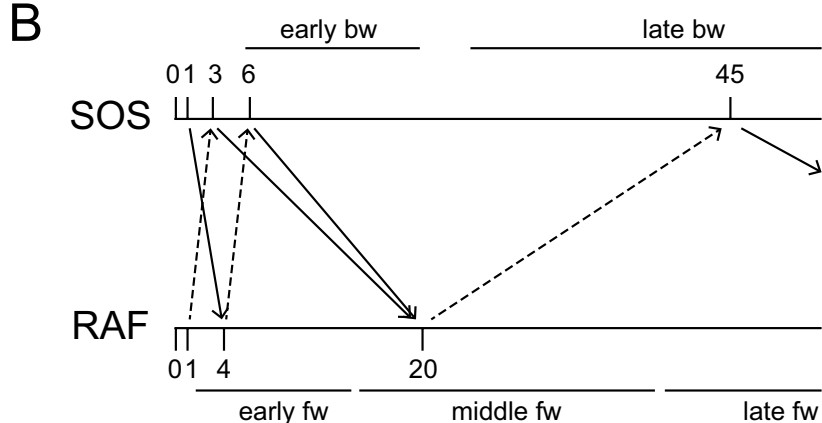

time after EGF stimulation (min)

**Appendix 1—figure 7.** TE exchange between SOS and RAF. (**A**) Time courses of TE were calculated for the SOS and RAF response using the total 1317 reaction dynamics data obtained under 0.1–100 ng/ml EGF (*Figure 1D*). The calculation started from Tc = 1, and the time indicated by the arrows in the obtained TE time courses was used for Tc in the next round of calculation in the reverse direction. (**B**) Diagram of TE exchange between SOS and RAF. Periods of the typical TE peaks are indicated.

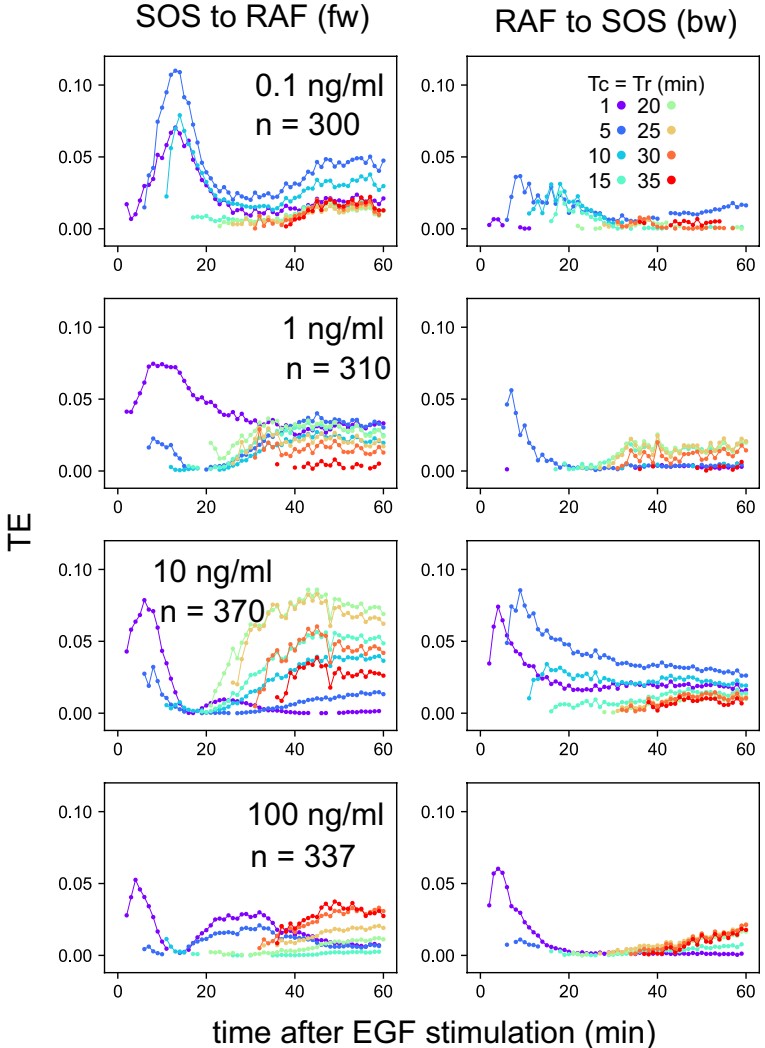

**Appendix 1—figure 8.** EGF dose dependency in the TE time courses. TE time courses were calculated for the SOS and RAF response time series at the indicated concentrations of EGF. n: data number. The calculations were performed for the time interval Tc = Tr from 1–35 min after the cell stimulation. Only statistically significant TE values (as defined in the Materials and methods section) are plotted.

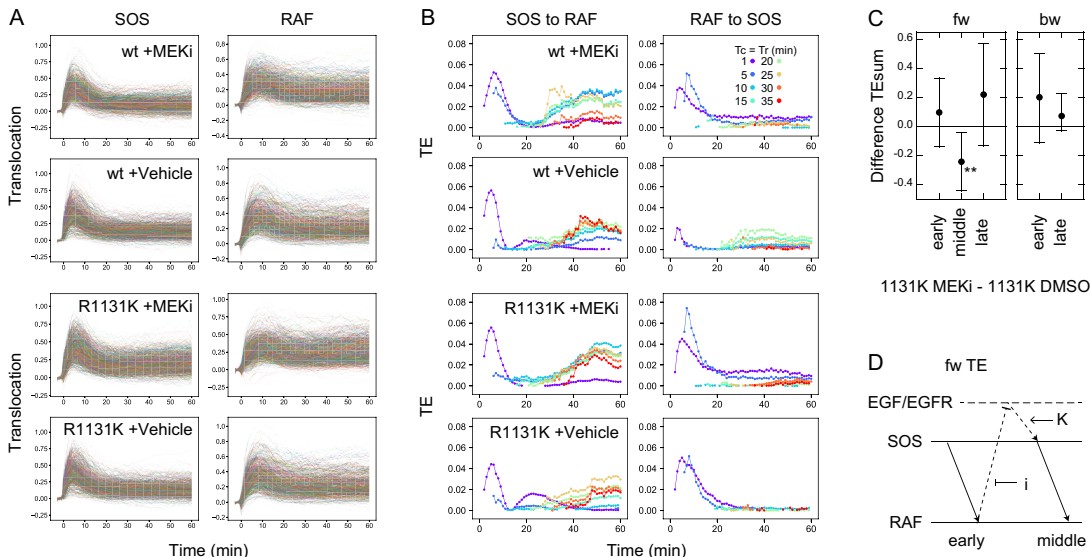

**Appendix 1—figure 9.** Effects of the MEK inhibitor in cells with a Noonan syndrome SOS. (**A**) Single-cell time courses of SOS and RAF responses for *Figure 5B*. (**B**) TE time courses for the data shown in (**A**). Calculations were performed for the indicated Tc = Tr from 1 to 35 min after cell stimulation. Only statistically significant TE values are plotted. (See Materials and methods.) (**C**) Difference between TEsum values in the presence and absence of the MEK inhibitor in R1131K cells. Positive values mean larger TEsum in the presence of the MEK inhibitor. Averages for 100 times bootstrapping are shown with the 5–95% interval. **bootstrapping p<0.05. (**D**) A possible pathway carrying middle fw-TE through a feedback loop. Action points of the MEK inhibitor (**i**) and R1131K mutation (**K**) are indicated.

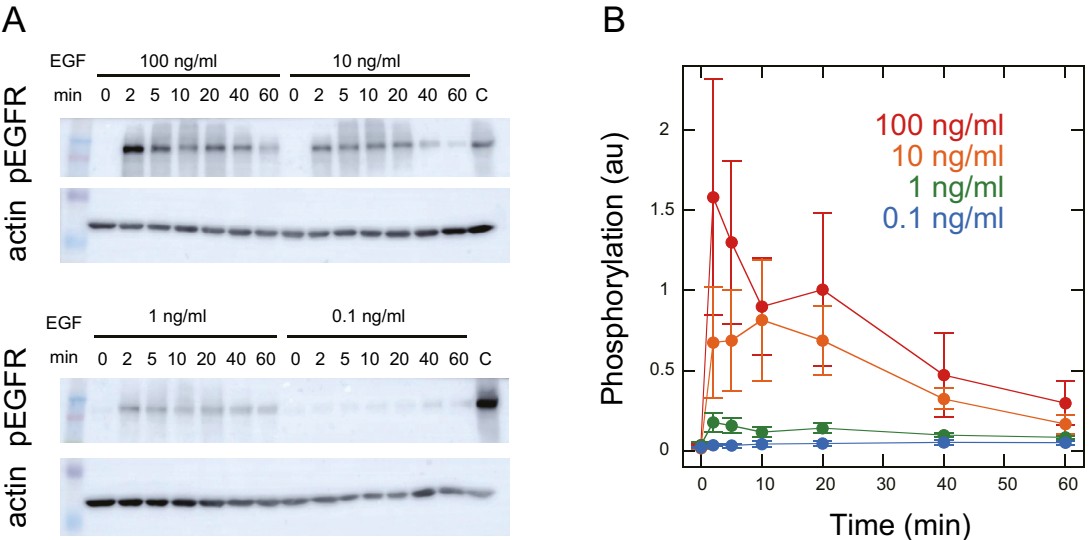

**Appendix 1—figure 10.** Time course of EGFR phosphorylation. Phosphorylation levels of Y1068 in EGFR indicating EGFR activity were measured. Representative Western blotting results (**A**) and quantification of four independent experiments (**B**) are shown. Cells expressing wt SOS and GFP-RAF were prepared under the same conditions as for SOS and RAF translocation measurements and stimulated with EGF. EGF dose and time after EGF stimulation are shown. Staining intensities were normalized to those for actin in the same sample and further to those of the same control sample of pEGFR (indicated by C). Phosphorylation levels peaked at 2–10 min after the EGF stimulation and decreased thereafter as expected from SOS, RAF response, and TE analysis.

The online version of this article includes the following source data for appendix 1—figure 10:

**Appendix 1—figure 10—source data 1.** Source data for *Appendix 1—figure 10*.

**Appendix 1—figure 10—source data 2.** Source data for *Appendix 1—figure 10*.

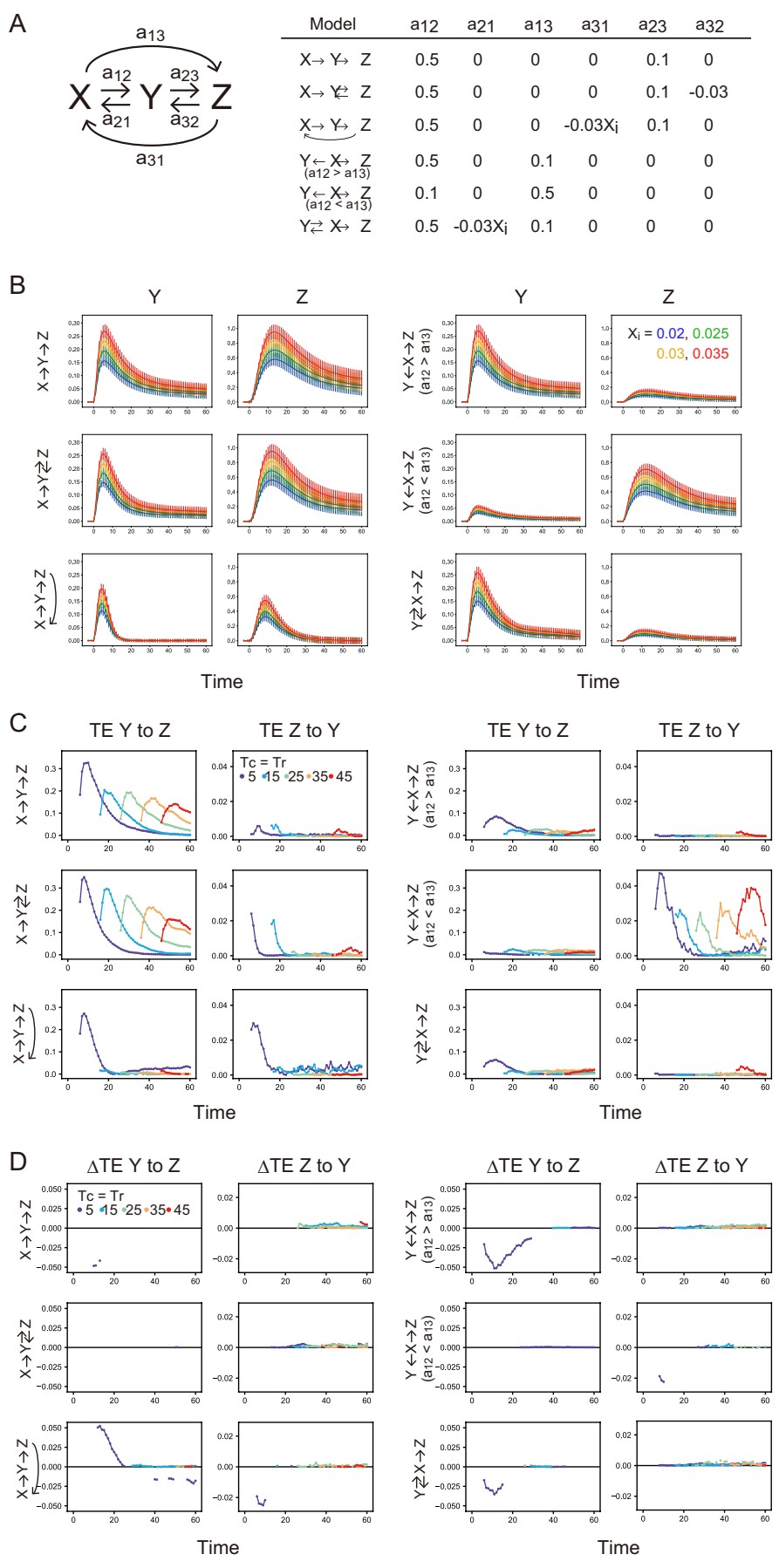

**Appendix 1—figure 11.** Stochastic simulation and TE calculation for simple reaction networks. (**A**) Network structure and reaction parameters. $X_i$ is the initial intensity of $X_0$ and controls the amplitude of X (see Supplement text). (**B**) Response time courses of X and Y. Averages of 500 simulated trajectories are shown with SD for each $X_i$ value. C. TE time courses for the mixed data under all $X_i$ values. (**D**) Differences between conditional and non-conditional TE to $X_i$. (**C, D**) The averages of 100 times bootstrap runs are shown. Only statistically significant values are plotted.

