## [Editor Report]

Intracellular signaling pathways and reaction networks are conventionally modeled analytically using ordinary differential equations. This paper demonstrates nicely how a statistical approach, transfer entropy, can be applied for model-free assessment of the timing, direction, and strength of the information flow regulating a system. Specifically, transfer entropy is applied to time-lapse imaging data to evaluate feedback loops in the EGFR-SOS-RAS-RAF pathway and the effects of drugs and mutations. Overall, the paper will be important for a wide audience including systems biologists and molecular pharmacologists, and is supported by compelling evidence.

---

## [Decision Letter]

[Editors' note: this paper was reviewed by Review Commons.]

---

## [Author Response]

General Statements

In addition to the point-by-point responses to the reviewer's comments, we have implemented the following changes to the manuscript to address errors in the previous version. We apologize for this inconvenience and wish to clarify that these changes do not affect the conclusion of the manuscript.

1. Figure S7A: ‘from RAF 6 min’ -> ‘from SOS 6 min, ‘from SOS 20 min’ -> ‘from RAF 20 min’, ‘from RAF 45 min’ -> ‘from SOS 45 min

2. Supplementary information, l. 191: ‘larger’ -> ‘smaller’, ‘~0.1’ -> ‘~0.01’.

Point-by-point description of the revisionsReviewer #1 (Evidence, reproducibility and clarity (Required)):This study applies calculations of transfer entropy (TE) to paired time series data on SOS and RAF translocation, collected from over 1000 individual cells under various conditions. This analysis allows the authors to analyze the timing and efficiency of information transfer between two steps of the EGFR signaling pathway. In addition to the expected correlation between SOS translocation, which is upstream of RAF translocation, the authors find a number of subtle phenomena in the analysis that indicate the presence of feedback structures, some of which are known, and others that may be novel. The study also examines the effects of a MEK inhibiting drug and a SOS mutation on the Major comments:Overall, the TE analysis appears to be clearly described and the authors provide reasonable justifications for the conclusions they draw, in terms of this analysis. What is more difficult to assess is the experimental reproducibility. While the main dataset analyzed contains hundreds of cells for each condition, it is not clear whether these are taken from a single experiment, where the many cells are responding in parallel to the same treatments, or if they represent independent runs of the experiment. If they are independent runs, can the separate runs be compared to assess run-to-run variation? How do the conclusions of the TE analysis vary run-to-run?More information should be included on how replicates were organized.

To show the reproducibility of the measurements, we added a two-dimensional plot of the single-cell response intensity of SOS and RAF at three representative time points (10, 20, and 30 min) after EGF stimulation, along with a list of the number of cells measured on each day and inter-day fluctuations of the response averages (Figure S2). As shown in this figure, we conducted a 6-day experiment for each EGF condition to obtain all the data sets shown in Figure 1. Each day, we had 20~100 cell measurements from 10~13 observation fields (l.125, 126; l.443, 444). Single-day distributions overlapped with a small bias (Figure S2A). Average daily variation was less than 30% of the total cellular response range for most measurements (Figure S2C). The data number per day was too small to calculate TE for evaluation of the reproducibility of the TE amount.

Some important controls seem to be missing. For example, how does the TE perform when non-translocating markers (such as GFP alone, or mutant forms of SOS/RAF that lack their recruitment domains) are used rather than the signaling molecules? Does RAF show translocation due to unlabeled SOS2 in the SOS1 knockout cells that do not have the SOS rescue construct? These are important to assess whether the information picked up by the TE analysis represents purely the observed molecules, or contributions from unlabeled processes.

There were no significant changes in the cell surface density of SOS and RAF after vehicle stimulation (l.127, 128; Figure 1D, black). This result may be difficult to see in the original figure, so we show it separately in Supplement Figure S3A. The result indicates that the translocations were specific for SOS and RAF proteins but were not due to the protein tags. Though the number of data (150) may not be enough, we conducted TE calculation and detected TEs one order smaller than those under EGF stimulations (l.146-148; Figure S3B). These small TEs may be noise in TE calculations (TE takes only zero or a positive value) or may mean actual information transfer between basal reactions in quiescent cells. Since no significant temporal structure was observed, it is unlikely that the TE was due to the effect of the medium application.

Because SOS2 would complement SOS1 in SOS1 knockout cells, it is difficult to assess SOS1 and SOS2 function separately. We expect that both SOS1 and SOS2 cause RAS activation leading to RAF translocation. The difference in information flow between wt SOS1 and R1131K SOS1 cells must be due to the R1131K mutation. This implies that R1131K SOS1 has a dominant-negative effect on SOS2 and that wt SOS1 at least partially regulates RAF translocation (l.376-379).

Minor commentsIt took me some time to find where the definitions of the early, middle, and late peaks were shown, until I found it in the table in 2D. It would be helpful to point this out more clearly when the peaks are first mentioned in the text.

We mention it at l.173, 174 in the revised text.

The sentence ending on line 337 with "primarily" seems to end abruptly and may be missing something.

The sentence was changed to “… that this regulation was primarily functional.” (l. 354)

Reviewer #1 (Significance (Required)):Overall, I think this paper demonstrates nicely how TE analysis can be applied to paired live-cell time series data to characterize the properties of a signaling pathway. The way in which the analysis is performed on single-cell pairs and integrated with existing knowledge of the pathway is quite sophisticated and suggests that such an approach could be of very significant usefulness, particularly in analyzing how drugs and mutations affect a signaling pathway.It is difficult to fully assess the generality of the analysis methods from just one pathway. Furthermore, many of the conclusions of the paper end with conjectures based on existing literature, but don't really test the ideas that come out of the analysis. While experimental tests could be fairly time-consuming in many of these cases, another possibility is to use data generated from simulations, where variations in feedback, etc. could be systematically tested (optional).

We added the results of TE analysis to the data produced by stochastic simulations. Please note that these simulations are not intended to reproduce phenomena in real cells, but to conceptually confirm TE behavior. The methods and results are described in the ‘Materials and methods’ section in the text and supplementary material ‘Stochastic ODE simulations’ section and Supplementary Figure S11. It has been confirmed that detecting a significant amount of bw-TE requires a feedback loop or parallel inputs (Supplementary text l.121-132), as expected in the text (l.308-311).

We also examined the difference between conditional and non-conditional TEs (∆TE; Figure S11D). As far as we examined, negative ∆TE was observed in the networks where the input X controls Y*_t_* or Z*_t_* in the fw or bw direction, respectively, independently of Y_Tr_ or Z_Tr,_ respectively, which is a necessary but not sufficient condition. Positive ∆TE was observed in a network containing a path from Y_Tc_ to Z*_t_* mediated by X (Y→Z→X→Y→Z). This is also a necessary condition (text l. 206-211, Supplement text l. 56-66; l. 133-242). These results partially support our expectations about the structure of the information network (Figure 3E). We did not observe a positive ∆TE in the networks shown in the right pathway of Figure 3Ea and the pathways in Figure 3b. However, we know only a necessary condition for positive ∆TE and considered only simple linear regulation in the simulations, so we cannot conclude that these pathway topologies are not used in the EGFR-RAS-MAPK system (Supplementary text l.133-142).

Advance: As the authors point out, the subtle changes that can be picked up by TE are likely much more informative than the more standard approach of simply asking how these drugs and mutations affect the downstream signaling magnitude.Audience: I think this is a very interesting contribution that will appeal to an audience with specific interests in EGFR signaling, but also those more generally interested in signal transduction or in other pathways.Expertise: EGFR signaling pathway, live-cell signaling methods, computational pathway modelingReviewer #2 (Evidence, reproducibility and clarity (Required)):In their manuscript, Umeki et al. apply transfer information on time-lapse signaling data of SOS and RAF to infer topological information and signaling flow of the EGFR pathway. They use life cell imaging of fluorescently labeled SOS and RAF to record the dynamics of receptor proximal EGFR signalling, and then apply a number of different metrics building on transfer entropy to disentangle signaling flow.In terms of data, there seems to be some overlap with Imaizumi et al. If this is so (not sure), this should be indicated.

Reaction trajectories in the cells expressing wt SOS under 100 ng/ml EGF without MEK inhibitor were taken in the same condition as in Imaizumi et al. (2022), but in the independent measurements. We mentioned this in the ‘Materials and Method’ section (l.445-447). All other measurements were performed under new conditions (EGF dose in the R1131K mutant experiment was different from that in Imaizumi et al.).

Overall, I find the approach interesting and potentially able to unravel signaling topologies. The manuscript is very difficult to read, particularly for a non-specialist in information theory it may be not appropriate. It would benefit from a rewrite. If this is for an trans- or interdisciplinary audience, the different concepts of how transfer energy is used need to be (better) described. For instance, conditional TE is not introduced and the reader would need an introduction to understand how it differs from TE.

We have added more detailed explanation of TE and conditional TE in the supplementary text (l.70, 71, in the text and ‘TE and conditional TE’ section in the Supplementary text). We have also added brief introduction to the use of TE analysis in the previous works (l.78-82 in the text and ‘TE and conditional TE’ section in the Supplementary text).

In the current form of the manuscript, it is difficult to judge the quality of the analysis and if the conclusions are warranted, as the manuscript lacks methodological detail. These missing details include the details of how the TE was computed and the associated random sampling and bootstrapping was performed, and how conditional is defined TR was computed. Also the precise definition of how TEsum was computed remains unclear. As these details matter and may influence if and how estimation bias influences the TE, these need to be included to fully judge the results.

We have added a figure to explain the procedure of TE calculation (l.480, 481, Figure S1).

We also added an explanation of the random sampling and bootstrapping (l.490-493).

The definition and calculation method of the conditional TE is written in lines 197-199 of the text. The definition of TEsum is written in lines 171-173 of the text.

Detailed comments: Figure 1 C+D: These panels are so small that it one cannot really see what is shown.

We have enlarged Figure 1C and D.

Page 7, line 185: conditional TEsum is not defined

We divided the sentence into two to make clear the definition of conditional TEsum (l.197-199). Conditional TEsum is the expected value of TEsum over the EGF concentrations (Supplementary text ‘TE and conditional TE’).

Page 7, line 211: we expected.… would be within.…. Why did the authors expect this?

The accuracy of the TE calculation depends on the number of data, which determines the accuracy of estimating the population distribution. We examined the effect of the number of data by changing the number of data set in the bootstrap (Figure S6). For more than 150 datasets (as in Figure 4), the TEsum value estimated as the mean of the bootstrap was in the range of 83-104% of the TEsum estimated by the entire 1317 datasets, except for the estimate for the early fw peak. For the early fw peak, the bootstrap averages were within the 63–107% range. Since the average TEsum converges to the average TEsum of the entire dataset as the number of data increases, we expected the converged value to be sufficiently accurate and used the range of bootstrap averages for the 150 data as a measure of the error range (l.175-177; l.225-228).

Page 9, lines 262-275: The description of the implementation of the logic gates is so convoluted that I cannot follow it. Also, I have not heared about ANDN gates. Please explain (or is this a typo?)

ANDN gate is described in de Ronde et al. (Biophys J. 103:1097, 2012). However, since an analogy to the logic gates is not necessary to explain the results of this study, we have omitted the analogy to reduce complexity.

Figure 4, x-axes of B,C, and Tc in A is inconsistent: Is it SOS 30 min or SOS 25min? Is it RAF 30 min or RAF 25?

We have corrected the mistake. SOS 30 min and RAF 25 min are correct. Thank you for pointed it out.

Referees cross-commentingI think this comment of reviewer 3 is very important (and I overlooked this problem):"The transfer entropy calculation assumed that the joint distribution of the variable is multivariate Gaussian. Is this assumption feasible? Particularly, the analysis was performed on a dataset comprising multiple conditions of EGF doses, which seem to adhere to multimodal distribution rather than the Gaussian distribution."

As examples, we show 2-dimensional projections of the 3-dimensional intensity distributions of (X(Tc), Y(Tr), Y(*t*)) at the representative time points in the TE peaks (X, Y are SOS and RAF response intensities, respectively; Figure S4). The distributions are unimodal but not multimodal. Most of their main bodies are Gaussian-like with little asymmetry, although the tails of the distributions are often skewed. The assumption of a Gaussian distribution is a first approximation (l.137-139), and this approximation was adopted because it is almost the only practical way to achieve detailed TE calculations from this type of experimental data. Furthermore, even in non-Gaussian cases, it is known that non-zero TE values under the Gaussian approximation mean non-zero values of the true TE (Marinazzo et al. Phys Rev Lett. 100; 144103, 2008) (l.481-484).

*Lines 158-169: This paragraph explains the proposed phase-dependent information flow based on the conditional TE analysis, as illustrated in Figure 3C. However, the justification of two key aspects remains unclear. Firstly, it is not evident how the existence of the feedback loop between SOS and RAF was established. Secondly, the rationale behind the conclusion that EGF input increases the forward information flow from SOS to RAF in the early phase needs elaboration. Please clarify these points to strengthen the argument for the proposed information flow.*

More specifically, the detection of a significant level of TE from RAF to SOS suggests two possibilities: feedback from RAF to SOS and/or parallel inputs from a hidden state to both SOS and RAF (as shown in Figure 3Eb). Based on prior knowledge, that several feedback regulations have been reported from RAF (l.312-319), it is likely that the feedback loops produced the bw-TE. Our result that the MEK inhibitor affected the amount of early bw-TE (Figure 5) supports this likelihood. To make clear these points, we have rewritten and added several sentences (l.148-151; l.308-311; l.319,320).

We did not say that EGF input increases the forward information flow in the submitted manuscript. The reviewer may be reading the old version that appeared on bioRxiv. (This might be related to the comment on Figures 2 and 3.) We have completely renewed the analysis method from the old version to the submitted one (current version in bioRxiv) to see the properties of TE in transient responses such as EGF-RAS-MAPK response. In the old method, we scanned Tc and Tr with *t* in TE calculations, keeping the time differences (*t*–Tc and *t*–Tr) constant. This method is usually used in the papers using TE analysis and is good for studying information flow in steady states. In the new method used in this version, we have fixed Tc and Tr in the TE calculations, scanning only *t* (Figure 2A). With this method, we are able to capture to which time (*t*) TE comes from a fixed time Tc. The relationship between the results of these two analyses is complicated and we do not want to say anything about it at this time. It would be too confusing for the reader.

Anyway, we could say that EGF/EGFR increased fw-TE, because TE was greatly increased by EGF stimulation (comparison between Figure S3 and Figure 2B). The result of conditional TE analysis indicates that EGF/EGFR was involved in the regulation of fw-TE levels. The larger conditional TE than the non-conditional one means that fw-TE was increased by specifying EGF/EGFR activity. These two results may be observing different aspects of the EGF/EGFR function in the information flow. We want to put this discussion on hold for now. It would require many experiments on the mechanism of the TE pathway.

Line 346: The transfer entropy calculation assumed that the joint distribution of the variable is multivariate Gaussian. Is this assumption feasible? Particularly, the analysis was performed on a dataset comprising multiple conditions of EGF doses, which seem to adhere to multimodal distribution rather than the Gaussian distribution. A statistical test is needed to determine if the data indeed follows a Gaussian distribution.

As examples, we show 2-dimensional projections of the 3-dimensional intensity distributions of (X(Tc), Y(Tr), Y(*t*)) at the representative time points in the TE peaks (X, Y are SOS and RAF response intensities, respectively; Figure S4). The distributions are unimodal but not multimodal. Most of their main bodies are Gaussian-like with little asymmetry, although the tails of the distributions are often skewed. The assumption of a Gaussian distribution is a first approximation (l.137-139), and this approximation was adopted because it is almost the only practical way to achieve detailed TE calculations from this type of experimental data. Furthermore, even in non-Gaussian cases, it is known that non-zero TE values under the Gaussian approximation mean non-zero values of the true TE (Marinazzo et al. Phys Rev Lett. 100; 144103, 2008) (l.481-484).

*Lines 166-169: The study concluded that EGF increases the forward information flow from SOS to RAF in the early phase, while direct regulation by EGF becomes negligible, and the feedback loop autonomously drives in the late phase. Does this imply that the information flow in the late phase is similar to that between SOS and RAF after the vehicle treatment? Please compare the information flow of the late phase and after the vehicle treatment.*

The TE time courses in cells with vehicle stimulation are shown in Figure S3. As shown, the late phase TE after the vehicle stimulation was much smaller than that after the EGF stimulation in the late phase. The greater TE under EGF stimulation must be related to the response time courses of SOS and RAF that were not returned to baseline during the 60 min of EGF stimulation (Figure 1D). The phosphorylation (activation) level of EGFR (Figure S10) and other proteins in the EGFR-RAS-MAPK system (Yoshizawa et al. Mol. Biol. Cell 32:1838, 2021; Okada et al. Biophys. J. 121:470, 2022) also does not return to the resting state after 60 min.

*Line 238: To further substantiate the proposed decrease in EGF regulation after the early phase (around 20 minutes), investigating a potential correlation between the duration of this early phase and the timescale of EGFR inactivation would be valuable. This could involve comparing the observed 20-minute window with established data on EGFR deactivation kinetics.*

We measured the time course of EGFR phosphorylation after EGF stimulation (Figure S10). The phosphorylation level at Y1068 in EGFR, which indicates EGFR activation, was peaked at 3-10 min after EGF stimulation and decreased after that. The result confirmed those we have reported before (Yoshizawa et al. Biophys. Physicobiol. 18:1, 2021; Okada et al. Biophys. J. 121:470, 2022).

The image resolution is too poor. In particular, the statistically significant levels of TE were denoted by red arrows in Figures 2 and 3, but these are too small to be clearly visible.

We are sorry, but do not understand which figure the reviewer is referring to in this comment. Figures 2 and 3 do not use red arrows. We have used 300 dpi jpeg format for the submitted figures, but the resolution may not be sufficient. Figures 2 and 3 in eps format will be submitted at the time of publication.

*Figure 4D. What is the definition of the response? How can it have a negative value?*

In Figure 4, we quantified the SOS and RAF responses as the integration of the response time profiles (Figure 1D) during the periods of TE peaks (l.679, 680). Because protein responses were measured as the changes in the molecular density from the basal value at time 0 (l.456, 457; l.638, 639), they can be negative, meaning decreases in protein density form the basal state.

*Figure S1. What is the definition of Tr and Tc? Please explain how the results shown in Figure S1 support the validity and significance of the TE calculations.*

In Figure S5 (Figure S1 in the submitted version), Tc and Tr indicate the time after EGF stimulation at which the response intensity distributions were taken for the mutual information calculation. The same Tr and Tc were used for TE calculations. We have clarified this definition (Supplement l.186, 187).

TE is the difference in mutual information as shown in Figure 2A. However, since subtraction is not explicitly performed in the TE calculation (equation in l.477), we cannot distinguish whether the estimated TE values are the errors of potential subtraction between similar values, appearing as a positive TE bias or not. In Figure S5, we examined the error values of each term of TE as the significance threshold in the bootstrapping. The errors were significantly smaller than the estimated TE values, indicating that the TE values were not determined by this type of errors. We have added explanations to the legend of Figure S5. At the suggestion of reviewer #2, we have added an introduction to mutual information and TE in the supplementary text, which will be helpful in clarifying the background of this investigation.

Reviewer #3 (Significance (Required)):The manuscript titled "Evaluation of information flows in the RAS-MAPK system using transfer entropy measurements" examines the information flow between SOS and RAF time series during signal transduction induced by EGF stimulation using transfer entropy. Through the analysis, context-specific information exchanges among the components are unveiled. Moreover, it is observed that the administration of an MEK inhibitor or the expression of an SOS mutant results in heightened information flow from RAF to SOS, consequently diminishing the ability of RAF to accurately respond to EGF inputs. Although the manuscript introduces significant novelty, there is potential to enhance its readability and clarity.